# Interpretable Decision Tree Search as a Markov Decision Process

## Abstract

Finding an optimal decision tree for a supervised learning task is a challenging combinatorial problem to solve at scale. It was recently proposed to frame this problem as a Markov Decision Problem (MDP) and use deep reinforcement learning to tackle scaling. Unfortunately, these methods are not competitive with the current branch-and-bound state of the art. Instead, we propose to scale the resolution of such MDPs using an information-theoretic *tests generating function* that heuristically, and dynamically for every state, limits the set of admissible test actions to a few good candidates. As a solver, we show empirically that our algorithm is at the very least competitive with branch-and-bound alternatives. As a machine learning tool, a key advantage of our approach is to solve for multiple complexity-performance trade-offs at virtually no additional cost. With such a set of solutions, a user can then select the tree that generalizes best and which has the interpretability level that best suits their needs, which no current branch-and-bound method allows.

## 1 Introduction

Decision trees (DTs) remain the dominant machine learning model in applications where interpretability is essential [Costa and Pedreira, 2023]. Thanks to recent advances in hardware, a new class of decision tree learning algorithms returning optimal trees has emerged [Bertsimas and Dunn, 2017, Demirovic et al., 2022, Mazumder et al., 2022]. These algorithms are based on a branch-and-bound solver that minimizes a regularized empirical loss, where the number of nodes is used as a regularizer. These optimization problems have long been known to be NP-Hard [Hyafil and Rivest, 1976] and despite hardware improvements, solvers of such problems do not scale well beyond trees of depth 3 when attributes take continuous values [Mazumder et al., 2022]. On the other hand, greedy approaches such as CART [Breiman et al., 1984] are still considered state-of-the-art decision tree algorithms because they scale and offer more advanced mechanisms to control the complexity of the tree. By framing decision tree learning as a sequential decision problem, and by carefully controlling the size of the search space, we achieve in this paper a best of both worlds, solving the combinatorial optimization problem with accuracies close to optimal ones, while improving scaling and offering a better control of the complexity-performance trade-off than any existing optimal algorithm.

To do so, we formulate the problem of decision tree learning as a Markov Decision Problem (MDP, [L. Puterman, 1994]) for which the optimal policy builds a decision tree. Actions in such an MDP include tests comparing an attribute to a threshold (a.k.a. splits). This action space could include *all* possible splits or a heuristically chosen subset, yielding a continuum between optimal algorithms and heuristic approaches. Furthermore, the reward function of the MDP encodes a trade-off between the complexity and the performance of the learned tree. In our work, complexity takes the meaning of simulatability [Lipton, 2018], i.e. the average number of splits the tree will perform on the train dataset. The MDP reward is parameterized by $\alpha$, trading-off between train accuracy and regularization. One

of the main benefits of our formulation is that the biggest share of the computational cost is due to the construction of the MDP transition function which is completely independent of $\alpha$, allowing us to find optimal policies for a large choice of values of $\alpha$ at virtually no additional cost.

Branch-and-Bound (BnB) algorithms similarly optimize a complexity performance trade-off [Demirovic et al., 2022, Mazumder et al., 2022] but require the user to provide the maximum number of test nodes as an input to their algorithm. Providing such a value a priori is difficult since a smaller tree (e.g. with 3 test nodes) might be only marginally worse on a given dataset than a larger tree (e.g. with 15 test nodes) with respect to the training accuracy but might generalize better or be deemed more interpretable a posteriori by the user. As such, it is critical to consider the multi-objective nature of the optimization problem and seek algorithms returning a set of trees that are located on the Pareto front of the complexity-performance trade-off. To the best of our knowledge, this has been so far neglected by BnB approaches. *None of the BnB implementations return a set of trees for different regularizer weights* unlike greedy algorithms like CART or C4.5 that can return trees with different complexity-performance trade-offs using minimal complexity post-pruning [Breiman et al., 1984], making it a more useful machine learning tool in practice.

## 2 Related Work

### 2.1 Optimal Decision Trees.

Decision tree learning has been formulated as an optimization problem in which the goal is to construct a tree that correctly fits the data while using a minimal number of splits. In [Bertsimas and Dunn, 2017, Aghaei et al., 2020, Verwer and Zhang, 2019], decision tree learning is formulated as a Mixed Integer Program (MIP). Instead of using a generic MIP solver, [Demirovic et al., 2022, Mazumder et al., 2022] design specialized solvers based on the Branch-and-Bound (BnB) principle. Quant-BnB [Mazumder et al., 2022] is currently the latest work in this line of research for datasets with continuous attributes and is considered state-of-the-art. However, direct optimization is not a convenient approach since finding the optimal tree is known to be NP-Hard [Hyafil and Rivest, 1976]. Despite hardware improvements, Quant-BnB does not scale beyond trees depth of 3. To reduce the search space, optimal decision tree algorithms on binary datasets, such as MurTree, Blossom and Pystreed [Demirovic et al., 2022, Demirović et al., 2023, van der Linden et al., 2023], employ heuristics to binarize a dataset with continuous attributes during a pre-processing step following for example the Minimum Description Length Principle [Rissanen, 1978]. The tests generating function of our MDP formulation is similar in principle except that it is state-dependent, which, as demonstrated experimentally, greatly improves the performance of our solver.

### 2.2 Greedy approaches.

Greedy approaches like CART iteratively partition the training dataset by taking the most informative splits in the sense of the Gini index or the entropy gain. CART is only one-step optimal but can scale to very deep trees. This might lead to overfitting and algorithms such as Minimal Complexity Post-Pruning (see Section 3.3 from [Breiman et al., 1984]) iteratively prune the deep tree, returning a set of smaller trees with decreasing complexity and potentially improved generalization. The trees returned by our algorithms provably dominate—in the multi-objective optimization sense—all the above smaller trees in terms of train accuracy vs. average number of tests performed, and we experimentally show that they often generalize better than the trees returned by CART.

### 2.3 Markov Decision Problem formulations.

In [Topin et al., 2021], a base MDP is extended to an Iterative Bounding MDP (IBMDP) allowing the use of any Deep Reinforcement Learning (DRL) algorithm to learn DT policies solving the base MDP. While more general and scalable, this method is not state-of-the-art for learning DTs for supervised learning tasks. Prior to IBMDPs, [Garlapati et al., 2015] formulated the learning of DTs for classification tasks with ordinal attributes as an MDP. To be able to handle continuous features, [Nunes et al., 2020] used Monte-Carlo tree search [Kocsis and Szepesvári, 2006] in combination with a tests generating function that limits the branching factor of the tree. Our MDP formulation is different as it considers a regularized objective while [Nunes et al., 2020] optimize accuracy on a validation set. Our tests generating function is also different and dramatically improves scaling as

shown in the comparison of Sec. 5.1.1, making our algorithm competitive with BnB solvers, while [Nunes et al., 2020] only compared their algorithm against greedy approaches. A comparison of our method with other MDP approaches is presented in the supplementary material.

### 2.4 Interpretability of Decision Trees.

The interpretability of a decision tree is usually associated with its complexity, e.g. its depth or its total number of nodes. For trees with 3 to 12 leaves, [Piltaver et al., 2016] observed a strong negative correlation between the number of leaves in a tree and a "comprehensibility" score given by users. Most of the literature considers the total number of test nodes as its complexity measure, but other definitions of complexity exist. [Lipton, 2018] coined the term *simulatability*, which is related to the average number of tests performed before taking a decision. This quantity naturally arises in our MDP formulation. We show in a qualitative study that both criteria are often correlated but on some datasets, DPDT returns an unbalanced tree with more test nodes that are only traversed by a few samples.

## 3 Decision Trees for Supervised Learning

Let us consider a training dataset $\mathcal{D} = \{(x_i, y_i)\}_{i \in \{1,...,N\}}$, made of (data, label) pairs, $(x_i, y_i) \in (X, Y)$, where $X \subseteq \mathbb{R}^p$. A decision tree $T$ sequentially applies tests to $x_i \in X$ before assigning it a value in $Y$, which we denote $T(x_i) \in Y$. The tree has two types of nodes: test nodes that apply a test and leaf nodes that assign a value in $Y$. A test compares the value of an attribute with a given threshold value, $x_{.,2} \leq 3$". In this paper, we focus on binary decision trees, where decision nodes split into a left and a right child with axis aligned splits as in [Breiman et al., 1984]. However, all our results generalize straitghforwardly to tests involving functions of multiple attributes. Furthermore, we look for trees with a maximum depth $D$, where $D$ is the maximum number of tests a tree can apply to classify a single $x_i \in X$. We let $\mathcal{T}_D$ be the set of all binary decision trees of depth $\leq D$. Given a loss $\ell$ defined on $Y \times Y$ we look for trees in $\mathcal{T}_D$ satisfying

$$T^* = \underset{T \in \mathcal{T}_D}{\operatorname{argmin}} \, \mathcal{L}_\alpha(T), \tag{1}$$

$$= \underset{T \in \mathcal{T}_D}{\operatorname{argmin}} \, \frac{1}{N} \sum_{i=0}^{N} \ell(y_i, T(x_i)) + \alpha C(T), \tag{2}$$

where $C : \mathcal{T} \to \mathbb{R}$ is a function that quantifies the complexity of a tree. It could be the number of nodes as in [Mazumder et al., 2022]. In our work, we are interested in the expected number of tests a tree applies on any arbitrary data $x \in \mathcal{D}$. As for $\ell$, in a regression problem $Y \subset \mathbb{R}$ and $\ell(y_i, T(x_i))$ can be $(y_i - T(x_i))^2$. For supervised classification problems, $Y = \{1, ..., K\}$, where $K$ is the number of class labels, and $\ell(y_i, T(x_i)) = \mathbb{1}_{\{y_i \neq T(x_i)\}}$. In our work, we focus on supervised classification but the MDP formulation extends naturally to regression.

## 4 Decision Tree Learning as an MDP

Our approach encodes the decision tree learning problem expressed by Eq. (2) as a finite horizon Markov Decision Problem (MDP) $\langle S, A, R_\alpha, P, D \rangle$. We present this MDP for a supervised classification problem with continuous features, but again, our method extends to regression and to other types of features. The state space of this MDP is made of subsets $X$ of the dataset $\mathcal{D}$ as well as a depth value $d$: $S = \{(X, d) \in P(\mathcal{D}) \times \{0, ..., D\}\}$, where $P(\mathcal{D})$ is the power set of $\mathcal{D}$. Let $\mathcal{F} = \{f : f(.) = \mathbb{1}_{\{. \leq x_{ij}\}}, \forall i \in \{1, ..., N\}, \forall j \in \{1, ..., p\}\}$ be a set of binary functions. We consider only tests that compare attributes to values within the dataset because comparing attributes to other values cannot further reduce the training objective. The action space $A$ of the MDP is then the set of all possible binary tests as well as class assignments: $A = \mathcal{F} \cup \{1, ..., K\}$. When taking an action $a \in \mathcal{F}$, the MDP will transit from state $(X, d)$ to either its "left" state $s_l = (X_l, d+1)$ or its "right" state $s_r = (X_r, d+1)$. In particular the MDP will transit to $s_l = (\{(x_i, y_i) \in X : a(x_i) = 1\}, d+1)$ with probability $p_l = \frac{|X_l|}{|X|}$ or to $s_r = (X \setminus X_l, d+1)$ with probability $p_r = 1 - p_l$. Furthermore, to enforce a maximum tree depth of $D$, whenever a state is $s = (., D)$ then only class assignment actions are possible in $s$. When taking an action in $\{1, ..., K\}$ the MDP will transit to a terminal state

denoted $s_{done}$ that is absorbing and has null rewards. The reward of taking an action $a$ in state $s$ is given by the parameterized mapping $R_\alpha : S \times A \to \mathbb{R}$ that enforces a trade-off between the expected number of tests and the classification accuracy. It is defined by:

$$R_\alpha(s, a) = R_\alpha((X, d), a),$$

$$= \begin{cases} -\alpha, & \text{if } a \in \mathcal{F}, \\ -\frac{1}{|X|} \sum_{y_i \in X} \mathbb{1}_{y_i \neq a} & \text{if } a \in \{1, ..., K\}. \end{cases}$$

The complexity-performance trade-off is encoded by the value $0 \leq \alpha \leq 1$, which is the price to pay to obtain more information by testing a feature. A more detailed study of the trade-off is given in section 6.4. The maximum depth parameter $D$ is a time horizon, i.e. the number of actions it is possible to take in one episode. An algorithm solving such an MDP can always return a deterministic policy [L. Puterman, 1994] of the form: $\pi : S \to A$ that maximizes the expected sum of rewards during an episode:

$$\pi = \operatorname*{argmax}_\pi J_\alpha(\pi), \tag{3}$$

$$J_\alpha(\pi) = \mathbb{E}\left[\sum_{t=0}^{D} R_\alpha(s_t, \pi(s_t))\right], \tag{4}$$

where the expectation is w.r.t. random variables $s_{t+1} \sim P(s_t, \pi(a_t))$ with initial state $s_0 = (\mathcal{D}, 0)$.

**From deterministic policy to binary DT.** One can transform any deterministic policy $\pi$ of the above MDP into a binary decision tree $T$ with a simple extraction routine $E(\pi, s)$, where $s \in S$ is a state. $E$ is defined recursively in the following manner. If $\pi(s)$ is a class assignment then $E(\pi, s)$ returns a leaf node with class assignment $\pi(s)$. Otherwise $E(\pi, s)$ returns a binary decision tree that has a test node $\pi(s)$ at its root, and $E(\pi, s_l)$ and $E(\pi, s_r)$ as, respectively, the left and right sub-trees of the root node. To obtain $T$ from $\pi$, we call $E(\pi, s_0)$ on the initial state $s_0 = (\mathcal{D}, 0)$.

**Equivalence of objectives.** When the complexity measure $C$ of $\mathcal{L}_\alpha$ is the expected number of tests performed by a decision tree, the key property of our MDP formulation is that finding the optimal policy in the MDP is equivalent to finding $T^*$, as given by the following proposition

**Proposition 1:** *Let $\pi$ be a deterministic policy of the MDP and $\pi^*$ one of its optimal deterministic policies, then $J_\alpha(\pi) = -\mathcal{L}_\alpha(E(\pi, s_0))$ and $T^* = E(\pi^*, s_0)$.*

The proof is given in the Appendix H.

# 5 Algorithm

We now present the Dynamic Programming Decision Tree (DPDT) algorithm. The algorithm is made of two essential steps. The first and most computationally expensive step constructs the MDP of Section 4. The second step is to solve it to obtain policies maximizing Eq.(4) for different values of $\alpha$. Both steps are now detailed.

## 5.1 Constructing the MDP

An algorithm constructing the MDP of Section 4 essentially computes the set of all possible decision trees of maximum depth $D$ whose decision nodes are in $\mathcal{F}$. This specific MDP is a directed acyclic graph. Each node of this graph corresponds to a state for which one computes the transition and reward functions. To limit memory usage of non-terminal nodes, instead of storing all the samples in $(X, d)$, we only store $d$ and the binary vector of size $N$, $x_{bin} = (\mathbb{1}_{\{x_i \in X\}})_{i \in \{1,...,N\}}$. Even then, considering all possible splits in $\mathcal{F}$ will not scale. We thus introduce a state-dependent action space $A_s$, much smaller than $A$ and populated by the tests generating function.

### 5.1.1 Tests generating functions

A tests generating function is any function $\phi$ of the form $\phi : S \to P(\mathcal{F})$, where $P(\mathcal{F})$ is the power set of all possible data splits $\mathcal{F}$. For a state $s \in S$, the state-dependent action space is defined by $A_s = \phi(s) \cup \{1, ..., K\}$. Because for a given state $s$ we might have that $\phi(s) \neq \mathcal{F}$, solving the

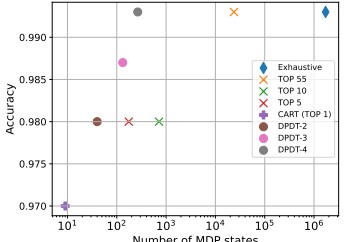

Figure 1: Comparison of DPDT algorithm on the Iris dataset in terms of the number of states in the MDP when using different tests generating functions. "TOP $B$" are tests function returning the $B$ most informative splits for each state. "Exhaustive" returns all possible states (equivalent to the search space of Quant-BnB). DPDT-$D_{cart}$ are the tests functions that make calls to the CART algorithm.

MDP with state-dependent actions $A_s$ is not guaranteed to yield the minimizing tree in Eq. (2), as optimization is now carried on a subset of $\mathcal{T}_D$. In this section, we compare different choices of $\phi$ on a sufficiently small dataset such that $\phi(s) = \mathcal{F}, \forall s \in S$ remains tractable. As a baseline, we use a tests generating function proposed in [Nunes et al., 2020], and compare with our proposed $\phi$ in terms of quality of the best tree vs. size of the MDP.

**Exhaustive function.** When $\phi(s) = \mathcal{F}, \forall s \in S$, the MDP contains all possible data splits. In this case, the MDP 'spans' all trees of depth at most $D$ and the solution to Eq. (4) will be the optimal decision tree of Eq. (2). In this case, the number of states in the MDP would be of the order of $\sum_{d=0}^{D-1} K(2Np)^d$ which scales exponentially with the maximum depth of the tree: this limits the learning to very shallow trees ($D \leq 3$) as discussed in [Mazumder et al., 2022]. The goal of a more heuristic choice of $\phi$ is to have a maximal number of splits $B = \max_{s \in S} |\phi(s)|$ that is orders of magnitude smaller than that of the exhaustive case $|\mathcal{F}| = Np$ such that the size of the MDP, which is now in the order of $\sum_{d=0}^{D-1} K(2B)^d$, remains tractable for deeper trees.

**Top $B$ most informative splits.** [Nunes et al., 2020] proposed to generate tests with a function that returns for any state $s = (X, d)$ the $B$ most informative splits over $X$ in the sense of entropy gain. In practice, we noticed that the returned set of splits lacked diversity

---

**Algorithm 1:** DPDT-$K$ MDP generation

**Data:** Dataset $\mathcal{D}$, max depth $D$
**Result:** Decision Tree Search MDP
$d \leftarrow 0$
$s_0 \leftarrow [\mathcal{D}, d]$
MDP.**AddState**$(s_0)$ # MDP of Sec. 4
**while** $d < D$ **do**
    # For all states at the current depth $d$
    **for** $s = (\mathcal{D}_s, d_s) \in MDP$ s.t. $d_s = d$ **do**
        # Test generating function follwing Sec. 5.1.1
        $T_{cart} \leftarrow$ **CART**$(\mathcal{D}_s, \text{maxdepth=K})$
        $A_s \leftarrow$ **ExtractSplits**$(T_{cart})$
        **for** $a \in A_s$ **do**
            # MDP expansion follwoing Sec. 4
            MDP.**AddRewardAndTransition**$(s, a)$
            MDP.**AddStates**(**NextStates**$(s, a)$)
        **end**
    **end**
    $d \leftarrow d + 1$;
**end**

---

and often consists of splits on the same attribute with minor changes to the threshold value. While this still leads to improvements over greedy methods—as shown in the study presented next—it is at the expense of a much larger MDP, i.e., search space.

**Top $B$ most discriminative splits.** Instead of returning the most informative splits, we propose at every state $s = (X, d)$, to find the most discriminative splits, i.e. the attribute comparisons with which one can best predict the class of data points in $X$. This is similar to the minimum description length principle used in [Demirovic et al., 2022] that transforms a dataset with continuous attributes to a binary dataset. However, we perform this transformation *dynamically* at every state while building the MDP. In practice, this amounts to calling CART with a maximum depth $D_{cart}$ (a hyperparameter of DPDT) on every state $s$, and using the test nodes of the tree returned by CART as $\phi(s)$.

While restricting the action space at a given state $s$ to the actions of the tests generating function $\phi(s)$ loses the guarantees of finding $T^*$, we are still guaranteed to find trees better than those of CART:

**Proposition 2:** *Let $\pi^*$ be an optimal deterministic policy of the MDP, where the action space at every state is restricted to the top $B$ most informative or discriminative splits. Let $T_0$ be the tree learned by CART and $\{T_1, \ldots, T_M\}$ be the set of trees returned by postprocessing pruning on $T_0$, then for any $\alpha > 0$, $\mathcal{L}_\alpha(E(\pi^*, s_0)) \leq \min_{0 \leq i \leq M} \mathcal{L}_\alpha(T_i)$.*

The proof for Prop. 5.1.1 follows from the fact that policies generating the tree returned by CART and all of its sub-trees (which is a superset of the trees returned by the pruning procedure) are representable in the MDP and by virtue of the optimality of $\pi^*$ and the equivalence in Prop. 4, are worse in terms of regularized loss $\mathcal{L}_\alpha$ than the tree $E(\pi^*, s_0)$. The consequences of Prop. 5.1.1 are clearly observed experimentally in Fig. 3. While this proposition holds for the latter two test generating functions, in practice, the tests returned by our proposed function are of much higher quality as discussed next.

**Comparing tests generating functions.** We conduct a small study comparing the exhaustive $\phi$ (labeled "Exhaustive") against the $\phi$ proposed in [Nunes et al., 2020] (labeled "Top B") and the one used in our algorithm (labeled "DPDT-K", where K is the maximum depth given to CART), on the Iris dataset. Figure 1 shows that while the latter two $\phi$ generalize the greedy approach (labeled "CART"), DPDT scales much more gracefully than when using the $\phi$ of [Nunes et al., 2020]. With $D_{cart} = 4$, DPDT-4 finds the optimal tree in an MDP having several orders of magnitude less states (a few hundreds vs a few millions) than the one built using the exhaustive $\phi$. This favorable comparison against exhaustive methods also holds for larger datasets as shown in Sec. 6.2.

The MDP construction of DPDT-K using the tests generating function is explained in Alg. 1. Starting from $s_0$, the state containing the whole dataset, CART with a maximum depth of K is called which generates a tree with up to $2^K - 1$ split nodes. These splits are what constitutes $A_{s_0}$, the set of binary tests admissible at $s_0$. For every such action, we compute the reward and transition probabilities to a set of new states at depth 1. This process is then iterated for every state at depth 1, calling CART with the same maximum depth of K on each of the states at depth 1, generating a new set of binary tests $A_s$ for each of these states $s$ and so on until reaching the maximum depth. Upon termination of Alg. 1, we compute the rewards for labelling actions at every state and we call the dynamic programming routine below to extract the optimal policy.

## 5.2 Dynamic Programming

Having built the MDP, we backpropagate using dynamic programming the best optimal actions from the terminal states to the initial states. We use Bellman's optimality equation to compute the value of the best actions recursively:

$$Q^*(s, a) = \mathbb{E}\left[r_{d+1} + \max_{a'} Q^*(s_{d+1}, a') | s_d = s, a_d = a\right],$$
$$= \sum_{s'} P(s, a, s')\left[R(s, a) + \max_{a'} Q^*(s', a')\right].$$

**Pareto front.** As our reward function is a linear combination of the complexity and performance measures, we can reach any tree "spanned" by the MDP that lies on the convex hull of the Pareto front of the complexity-performance trade-off. In DPDT, we compute the optimal policy for several choices of $\alpha$ using a vectorial representation of the $Q$-function that now depends on $\alpha$:

$$Q^*(s, a, \alpha) = \sum_{s'} P(s, a, s')\left[R_\alpha(s, a) + \max_{a'} Q^*(s', a', \alpha)\right].$$

We can then find all policies greedy w.r.t. $Q^*$ $\pi^*(s, \alpha) = \underset{a \in A}{\mathrm{argmax}} Q^*(s, a, \alpha)$. Such policies satisfy Eq. (4) for any value of $\alpha$. Given a set of values of $\alpha$ in $[0, 1]$, we can compute in a single backward pass $Q^*(s, a, \alpha)$ and $\pi^*(s, \alpha)$ and return a set of trees, optimal for different values of $\alpha$ (see Fig.7 for an illustrative example). In practice, the computational cost is dominated by the construction of the MDP 1 and one can promptly back-propagate the $Q$-values of over $10^3$ values of $\alpha$.

# 6 Experiments

In this section we study DPDT from different perspectives. First, in Sec. 6.2, we study DPDT in terms of its performance as a solver for the combinatorial optimization problem of Eq. (2). Here, we focus on smaller problems (maximum depth $\leq 3$) in which the optimal solution can be computed by Branch-and-Bound (BnB) algorithms. In this first set of experiments, we only report the training accuracy vs. the wall-clock time as done in prior work [Mazumder et al., 2022]. Then we study DPDT

for model selection (Sec. 6.3). From the perspective of the end user, a decision tree algorithm may be used for selecting either a tree that generalizes well to unseen data or a tree that is interpretable. We compare classification of unseen data of trees obtained by DPDT to other baselines described below. Then, we plot the train accuracy of trees learned by CART and DPDT as a function of their complexity to observe how a user can choose the complexity-performance trade-off. We use the 16 classification datasets with continuous attributes experimented with in [Mazumder et al., 2022].

When considering other optimal BnB baselines [Demirovic et al., 2022, van der Linden et al., 2023], two problems arise for fair comparison with DPDT in terms of model selection. First, to obtain a set of tree from such baselines, the optmization algorithms need to be ran as many times as trees wanted by the user. For example, one can obtain a set of trees of depth $\leq 5$ by running MurTree $2^5$ times with different maximum number of test nodes allowed in the learned trees. This could require up to $2^5$ times the runtime of a single optimization. Second, MurTree and `Pystreed` [Demirovic et al., 2022, van der Linden et al., 2023] require binary attributes. Learned trees are not comparable directly with trees trained on continuous attributes because each tree node testing a binary feature actually does at least two tests on the original continuous feature (see Appendix F.1 or Appendix D1 from [Mazumder et al., 2022]). DPDT is coded in Python and the code is available in the supplementary material. All experiments are run on a single core from a `Intel i7-8665U` CPU. All the links to code used for the baselines are given in the Appenix A

## 6.1 Baselines

**Quant-BnB.** [Mazumder et al., 2022] propose a scalable BnB algorithm that returns optimal trees. We emphasize that *Quant-BnB is not meant to scale beyond tree depths of 3* (explicitly stated in the Quant-BnB paper) and the authors' implementation of Quant-BnB does not support learning trees of depth $> 3$.

**MurTree,** `Pystreed`. To use [Demirovic et al., 2022, van der Linden et al., 2023] with continuous features datasets, the minimum length description principle is used to obtain bins in a continuous feature domain, then a one hot encoding is applied to binarize the binned dataset. This can result in datasets with more than $500$ features. As MurTree and `Pystreed` memory scales with the square of number of binary attributes, using those algorithms to find trees of depths greater than 3 often results in *Out Of Memory* (OOM) errors.

**Deep Reinforcement Learning.** We use Custard [Topin et al., 2021] as a DRL baseline. Custard has two hyperparameters: the DRL algorithm to learn a policy in the IBMDP and a tests generating function that gives p tests per feature. In our experiments, Custard-5 and Custard-3 correspond to DQN agents [Mnih et al., 2015] that can test each dataset attribute against 5 or 3 values respectively.

**CART** [Breiman et al., 1984] is a greedy algorithm that can build suboptimal trees for any dataset.

## 6.2 Optimality gap

Because we use a tests generating function that heuristically reduces the search space, a first question we want to investigate is how good is our solver for the combinatorial problem of decision tree search. To do so, we focus on max depth 3 problems for which $T^*$ can be computed exactly using Quant-BnB [Mazumder et al., 2022]. As Quant-BnB has a different complexity regularization (number of nodes in the tree) than DPDT (average number of tests per classified data), we set the complexity regularizing term $\alpha$ to 0 to allow direct comparisons. This does not create an artificial learning and on 14 out of 16 datasets, trees with $\alpha = 0$ generalize best, and second best on the remaining 2. That is because at depth 3 the risk of overfitting is small.

We run DPDT with calls to CART with maximum depth 4 or 5 as a tests generating function (DPDT-4 and DPDT-5 respectively). Quant-BnB is first run without a time limit to obtain optimal decision trees w.r.t. Eq.(2). Quant-BnB is also run a second time with a time limit equal to DPDT-5's runtime (we also added in the supplementary material results for Quant-BnB-$T$+5 and Quant-BnB-$T$+50 that add extra seconds to Quant-BnB-$T$). CART is run with the maximum depth set to 3 and the information gain based on entropy. All algorithms are run on the same hardware. Custard is run 5 times per dataset because it is a stochastic algorithm. We use stable-baselines3 implementation of DQN [Raffin et al., 2021] with default hyperparameters. A Custard run usually takes 10 minutes. We provide learning curves in Fig. 4. The key result from Table 1 is that DPDT-5 has better train accuracies than the

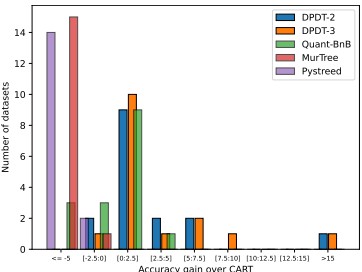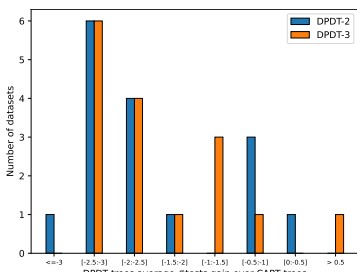

Figure 2: Performance gain of DTs over CART trees. Left, accuracy on unseen data gain of trees with depth ≤ 5 selected with procedure of Sec. 6.3. Right, average number of tests of those trees.

309 other non-greedy methods when run in similar runtimes across all classification tasks. Furthermore,
310 the train accuracy gaps between the optimal decision trees obtained from Quant-BnB, in sometimes
311 several hours, and DPDT are usually small (the maximum gap is 1.5% for the bean dataset).

Table 1: Train accuracy of decision tree algorithms. The "Quant-BnB" columns correspond to results for Quant-BnB with no time limit, i.e returing the optimal tree. The "Quant-BnB-$T$" column corresponds to results for Quant-BnB run for as long as DPDT-5. The "Greedy" columns correspond to CART with maximum depth of 3.

| Datasets | | | | Accuracy (train %) of depth-3 trees | | | | | | | Runtime in seconds | | | | | |
|---|---|---|---|---|---|---|---|---|---|---|---|---|---|---|---|---|
| Names | Samples | $p$ | Classes | Quant-BnB | Quant-BnB-$T$ | DPDT-5 | DPDT-4 | Custard-5 | Custard-3 | Greedy | Quant-BnB | DPDT-5 | DPDT-4 | Custard-5 | Custard-3 | Greedy |
| avila | 10430 | 10 | 12 | 58.5 | 57.3 | **58.5**\* | 58 | 40.9 ± 0.6 | 41 ± 0.3 | 53.8 | 4188 | 5.645 | **2.14** | 553 | 632 | 0.031 |
| bank | 1097 | 4 | 2 | 98.3 | 97.1 | **98** | **98** | 49.6 ± 1.6 | 35.5 ± 20.9 | 95.3 | 4.4 | 0.158 | **0.142** | 648 | 661 | 0.003 |
| bean | 10888 | 16 | 7 | 87.1 | 85.3 | **85.6** | 85 | 18.2 ± 2.3 | 19.2 ± 4.1 | 80.5 | 1014 | 16.194 | **5.836** | 697 | 687 | 0.114 |
| bidding | 5056 | 9 | 2 | 99.3 | 98.6 | **99.3**\* | **99.3**\* | 81 ± 4.4 | 79.4 ± 2.1 | 98.2 | 30 | 0.545 | **0.377** | 693 | 671 | 0.006 |
| eeg | 11984 | 14 | 2 | 70.8 | 68.3 | **70.3** | 70 | 54.9 ± 0.1 | 54.8 ± 0.5 | 66.6 | 4042 | 8.927 | **3.032** | 692 | 682 | 0.023 |
| fault | 1552 | 27 | 7 | 68.2 | 64.6 | **68** | 65.7 | 30.3 ± 1.4 | 27.5 ± 8.6 | 55.3 | – | 2.46 | **1.243** | 720 | 711 | 0.015 |
| htru | 14318 | 8 | 2 | 98.1 | **98** | **98** | **98** | 86 ± 0.9 | 59.4 ± 31.7 | 97.9 | 10303 | 11.316 | **4.246** | 690 | 684 | 0.055 |
| magic | 15216 | 10 | 2 | 83.1 | 82.6 | **83** | 82.7 | 58.1 ± 4.3 | 58.5 ± 3.0 | 79.3 | 1090 | 14.838 | **5.443** | 685 | 675 | 0.069 |
| occupancy | 8143 | 5 | 2 | 99.4 | 99.3 | **99.4**\* | 99.3 | 64.7 ± 0.5 | 65.1 ± 8.3 | 99.1 | 106 | 1.458 | **0.786** | 687 | 664 | 0.008 |
| page | 4378 | 10 | 5 | 97.1 | 96.5 | **97** | 97.0 | 90.2 ± 0.4 | 88.3 ± 4.8 | 96.3 | 471 | 2.859 | **1.29** | 708 | 687 | 0.01 |
| raisin | 720 | 7 | 2 | 89.4 | 88.1 | **88.5** | 88.3 | 50.9 ± 2.2 | 49.7 ± 1.0 | 86.9 | 167 | 0.501 | **0.3** | 668 | 667 | 0.003 |
| rice | 3048 | 7 | 2 | 93.8 | **93.7** | **93.7** | 93.6 | 51.9 ± 0.9 | 48.1 ± 3.4 | 93.0 | 1340 | 2.004 | **0.809** | 668 | 666 | 0.01 |
| room | 8103 | 16 | 4 | 99.2 | 98.8 | **99.2**\* | **99.2**\* | 71.5 ± 3.4 | 67.6 ± 5.6 | 97.7 | 180 | 2.714 | **1.884** | 1362 | 1389 | 0.01 |
| segment | 1848 | 18 | 7 | 88.7 | 79.1 | **88.2** | **88.2** | 13.7 ± 0.5 | 13.9 ± 0.5 | 81.6 | 153 | 0.771 | **0.397** | 812 | 761 | 0.009 |
| skin | 196045 | 3 | 2 | 96.9 | **96.7** | **96.7** | **96.7** | 61.2 ± 2.2 | 62.2 ± 8.7 | 96.6 | 350 | 48.894 | **19.239** | 752 | 745 | 0.082 |
| wilt | 4339 | 5 | 2 | 99.6 | 99.4 | **99.5** | **99.5** | 98.4 ± 0.2 | 98.3 ± 0.1 | 99.1 | 67 | 0.582 | **0.352** | 663 | 610 | 0.008 |

## 6.3 Selecting the best tree for unseen data

313 We now investigate whether DPDT is suited for model selection i.e. whether DPDT can identify an
314 accurate decision tree that will generalize well to unseen data for a given classification task. We used
315 the following model selection procedure for each classification task. First, we learn a *set* of decision
316 trees of depth $D \leq 5$ with DPDT-3, DPDT-2, and CART on a training set using different values of
317 $\alpha$ for DPDT or minimal complexity post-pruning for CART. Because Quant-BnB simply cannot
318 compute trees of depth $> 3$, we only report the accuracy on unseen data of Quant-BnB trees from
319 Table 1. Because the BnB baselines MurTree and Pystreed are not designed to return a set of trees,
320 we brute force the computation of at most $2^5$ trees from each by setting the maximum tree nodes
321 parameter to $0, ..., 2^5 - 1$. Then, for each baseline we evaluate each learned tree (only one tree for
322 Quant-BnB) on a test set and select the tree with highest test accuracy. Fig 2 reports the number of
323 datasets for which each baseline has better generalization performances than CART, and the number
324 of datasets for which DPDT-K returned trees performing less tests on average than CART trees. A
325 table with accuracies of the selected trees on a validation set, the runtime in seconds to obtain the set
326 of trees to select from, and the average number of tests performed on data in Appendix. All BnB
327 baselines required more than 5 minutes to generate a single tree. As such, the runtime for BnB to
328 obtain the whole set of trees is order of magnitudes higher than CART and DPDT. DPDT learns a set
329 of trees of at most depth 5 on the complexity-performance convex-hull in seconds which highlights
330 its ability to scale to non-shallow trees. For that purpose, DPDT built the MDP of possible solution
331 trees of at most depth 5 using CART as a tests generating function, and backpropagated state-action
332 values for 1000 different $\alpha$.

333 After applying the above selection procedure, we see on Table 2 that DPDT generalized better than
334 CART on 10 out of 16 datasets while CART outperformed DPDT on only one dataset. When accuracy
335 on test data for CART is already close to 100%, our approach can of course not largely outperform it.
336 However, the benefits of our method have to also be appreciated in terms of gains in average number

of tests. We can see that when CART does not generalize well, our method can have clear gains in generalization (e.g. avila, eeg and fault). Otherwise, when CART is close to 100% accuracy, our method can achieve similar results with less tests. In raisin, rice and room we need two fewer tests which is substantial when tests are expensive, e.g. an MRI scan when testing patients.

## 6.4   Selecting the most interpretable tree

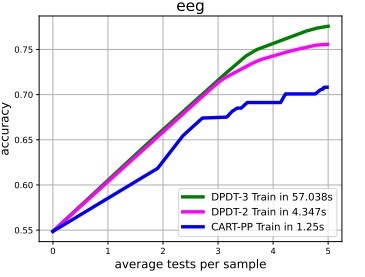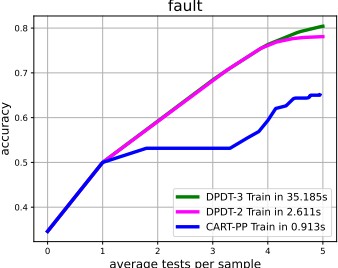

Figure 3: Complexity-performance trade-offs of CART and DPDT on two different classification datasets. CART returns a set of trees with the minimal complexity post-pruning algorithm. DPDT returns a set of trees by returning policies for 1000 different $\alpha$.

In this section, we show how a user can use DPDT to select a tree with complexity preferences. In Figure 3, we plot the trade-offs of trees returned by CART and DPDT. The trade-off is between accuracy and average number of tests. Because this is the trade-off that DPDT optimizes and because the trees of CART are "spanned" by the MDP created by DPDT, all trees returned by DPDT will dominate in the multi-objective sense trees returned by CART. This is well demonstrated in practice by Figure 3 where the curve of DPDT is always above that of CART. Learned trees and their accuracies as functions of number of nodes and tests are presented in Appendices 5 8 9. Finally, decision tree search being a combinatorial problem, there are always limits to scalability. In Appendix 6 we scale up to a tree depth of 10 by running DPDT-2 up to a depth of 6 then switch to DPTD-1 (i.e. greedy) thereafter. The rationale is that a non-greedy approach is more critical closer to the root.

# 7   Limitations, Future Work, and Conclusion

**Limitations.** In our opinion, both the strength and the weakness of DPDT come from the choice of the tests generating function. If the tests generating function generates too much tests in each MDP state, the runtime will grow and there is a risk for out-of-memory errors. This can be alleviated with parallelizing (expanding MDP states on different processes) and caching (only expand unseen MDP states), similar to [Demirovic et al., 2022]. A rule of thumb for running DPDT on personal CPUs is to choose a tests generating function resulting in an MDP with at most $10^6$ states.

**Future Work.** DPDT could scale to bigger datasets by combining Custard [Topin et al., 2021] with tests generating functions and tabular deep learning techniques [Kossen et al., 2021]. The latter is a promising research avenue. The transformer-based architecture from [Kossen et al., 2021] takes a *whole* train dataset as input and learns representations taking in account relationships between *all* training samples and *all* labels. Test actions are then the output of such a neural architecture: the tests generating function is *learned*.

**Conclusion.** In this work we solve MDPs whose optimal policies are decision trees optimizing a trade-off between tree accuracy and complexity. We introduced the Dynamic Programming Decision Tree algorithm that returns several optimal policies for different reward functions. DPDT has reasonable runtimes and is able to scale to trees with depth greater than 3 using information-theoretic tests generating functions. To the best of our knowledge, DPDT is the first scalable decision tree search algorithm that runs fast enough on continuous attributes to be an alternative to CART for model selection of any-depth trees. DPDT is a promising research avenue for new algorithms offering human users a greater control than CART over tree selection in terms of generalization performance and interpretability.

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

# A  Code links

**Quant-BnB.** The Julia code for Quant-BnB is available at https://github.com/mengxianglgal/Quant-BnB.

**CART.** We use the scikit-learn Cython implementation of CART available at https://scikit-learn.org/stable/modules/tree.html#tree-classification with the criterion parameter fixed to "entropy".

**MurTree,** Pystreed. Codes are available at https://github.com/MurTree/pymurtree and at https://github.com/AlgTUDelft/pystreed.

# B  On the failure of deep reinforcement learning.

For the dataset $X = \{(1, 2), (2, 1), (3, 4), (4, 3)\}$, $Y = \{0, 1, 2, 3\}$ both our MDP and IBMDP are equivalent for learning the optimal decision tree of depth 2. We show on Fig. 4 that two different DRL algorithms exhibit opposite performance: DQN can learn the optimal decision tree while PPO [Schulman et al., 2017] cannot. For that reason, we only trained Custard using DQN as the DRL agent. We see on Fig. 4 and Table 1 that Custard-5 *converged* to trees worst than CART for all classification datasets. This shows that while more scalable, DRL approaches are still not competitive on these types of problems. [Kohler et al., 2023] studied potential failure modes of DRL in our setting.

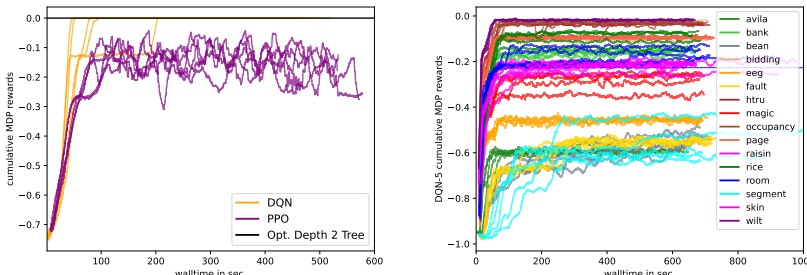

Figure 4: Left, DRL to learn the optimal depth 2 tree. Right, Custard-5 to learn depth 3 decision trees on classification datasets

## C Tree plots

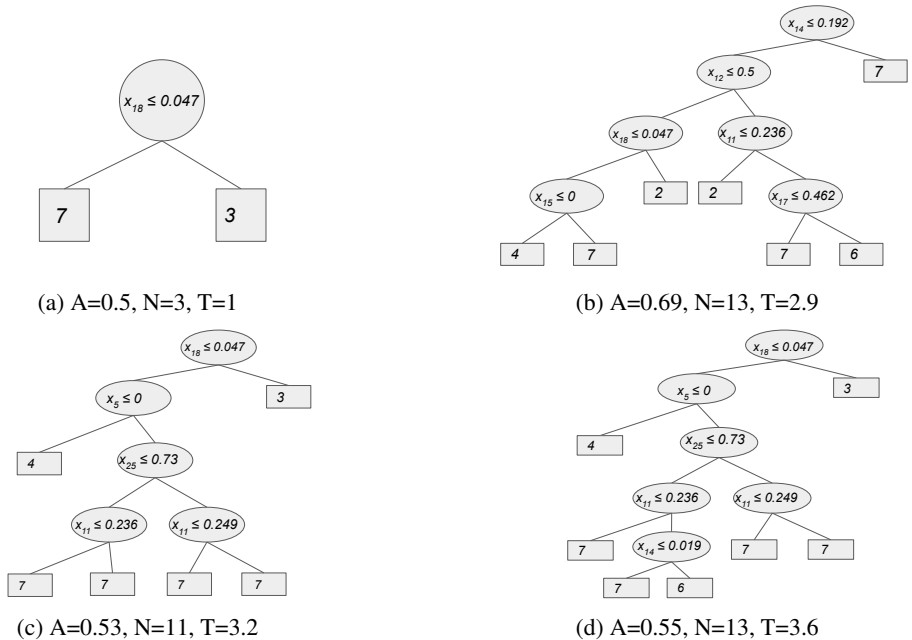

Figure 5: Trees for the fault dataset. Top: trees from DPDT. Bottom: trees from CART. A is accuracy, N the number of nodes, T the average number of tests.

## D Schematics DTDP

## E Detailed res of model selection

## F Comparisons with baselines operating on binary datasets

### F.1 Why comparisons with baselines that binarize datasets is not fair in our favor?

Algorithms finding optimal DTs for binary datasets such as MurTree [Demirovic et al., 2022] use a binarization method to transform a dataset with continuous attributes to a dataset with binary attributes. However, a DT learned on the binary dataset, whenever it tests the value of a binary attribute, can lead to up to two tests on the respective continuous attribute. Hence, DTs of a given maximum depth on the binary dataset are actually deeper if transformed into DTs on the original dataset with continuous attributes. Despite this, we show in Table 1 of this supplementary material

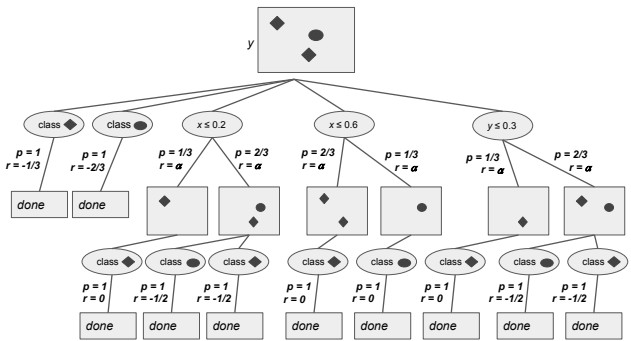

Figure 6: MDP for a training dataset made of three samples (illustrated with an oval and 2 diamonds), two continuous attributes ($x$ and $y$), and two classes. The tests generating function generated three possible tests. There is an initial state $(\mathcal{D}, 0)$ (the training dataset at depth 0), and six non-terminal states (three tests times two children states). Rewards are either $\alpha$ or the misclassification, and transition probabilities are one, or the size of the child state over the size of the parent.

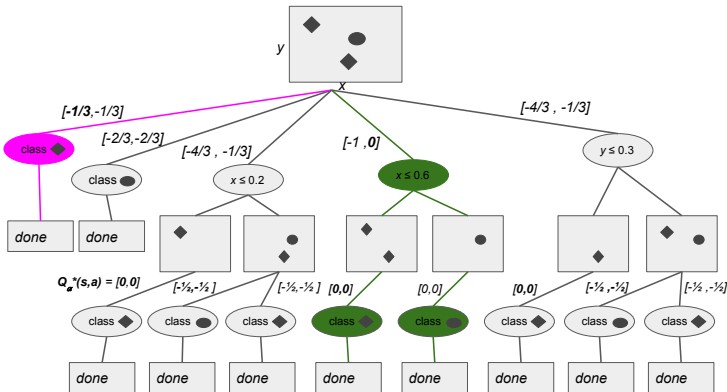

Figure 7: For $\alpha = 0$ and $\alpha = 1$, the values of $Q^*(s, a, \alpha)$ are backpropagated from leaf states to the initial state and are given in squared brackets. The optimal policy $\pi^*(., \alpha = 1)$, in pink, is a depth-0 tree with accuracy $\frac{2}{3}$. The optimal policy $\pi^*(., \alpha = 0)$, in green, is a depth-1 tree with accuracy 1.

Table 2: Trees of depth $\leq 5$ selected with the procedure described in Sec. 6.3.

| Datasets | Accuracy (%) on unseen data | | | | | | Runtime (s.) | | | Average Nb.Tests | | |
| Names | DPDT-3 | DPDT-2 | CART | Quant-BnB | MurTree | Pystreed | DPDT-3 | DPDT-2 | CART | DPDT-3 | DPDT-2 | CART |
|---|---|---|---|---|---|---|---|---|---|---|---|---|
| avila | **66.9** | 65.7 | 60.5 | 57.3 | OOM | OOM | 51.625 | 2.701 | 1.031 | 4.9 | 4.9 | **4.8** |
| bank | **99.3** | 97.8 | **99.3** | 97.8 | 48.6 | 48.6 | 2.054 | 0.353 | 0.031 | **3.2** | 3.7 | 3.4 |
| bean | **91.1** | **91.1** | 89.9 | 84.7 | OOM | OOM | 88.142 | 7.571 | 5.369 | **4.6** | 4.9 | 5.0 |
| bidding | **99.2** | **99.2** | **99.2** | 98.5 | 97.5 | 97.5 | 2.963 | 0.545 | 0.081 | **1.4** | **1.4** | 2.3 |
| eeg | **78.0** | 74.6 | 73.0 | 73.0 | OOM | OOM | 57.038 | 4.347 | 0.892 | **4.6** | 4.8 | 5.0 |
| fault | 71.8 | **72.8** | 57.9 | 61.2 | OOM | OOM | 35.185 | 2.611 | 0.536 | 5.0 | **4.5** | 4.9 |
| htru | 98.0 | **98.3** | **98.3** | 97.9 | OOM | 91.2 | 63.519 | 5.189 | 2.174 | **1.1** | 2.4 | 4.7 |
| magic | 84.5 | **84.8** | 82.5 | 82.1 | OOM | OOM | 98.623 | 7.06 | 3.189 | 5.0 | **4.8** | 5.0 |
| occupancy | **99.5** | **99.5** | **99.5** | 96.3 | OOM | 82.3 | 11.113 | 1.263 | 0.162 | **1.0** | **1.0** | 1.4 |
| page | **97.1** | **97.1** | 96.7 | 95.8 | OOM | 93.4 | 26.596 | 2.547 | 0.369 | 3.5 | 5.0 | 4.8 |
| raisin | 87.8 | **91.1** | 90.0 | 89.0 | 45.6 | 45.6 | 7.756 | 1.775 | 0.069 | 3.1 | **2.3** | 4.5 |
| rice | 93.7 | **94.2** | 93.4 | 93.9 | 87.1 | 87.1 | 17.915 | 1.693 | 0.356 | **1.6** | 1.7 | 3.6 |
| room | 99.2 | **99.4** | **99.4** | 98.6 | OOM | OOM | 19.134 | 1.574 | 0.247 | 2.5 | **2.3** | 4.1 |
| segment | **93.5** | 93.1 | 87.4 | 82.7 | OOM | OOM | 6.488 | 0.879 | 0.184 | **3.7** | 3.9 | 3.9 |
| skin | **99.5** | 99.2 | 98.6 | 98.6 | OOM | OOM | 265.243 | 18.066 | 1.985 | **3.8** | **3.8** | 4.2 |
| wilt | 87.2 | 84.8 | **87.6** | 81.3 | 70.4 | 70.4 | 3.898 | 0.462 | 0.125 | 4.1 | **3.2** | 3.9 |

that DPDT typically finds better solutions (in terms of training accuracy) than MurTree + binarization even though the comparison is not fair in our favor since MurTree is considering deeper trees.

To illustrate this unbalance with an example, we present a dataset with 3 samples, 2 classes, and 1 continuous attribute. After binning the continuous attribute and binarizing the dataset into 3 binary attributes, we compute the optimal depth 1 tree like [Demirovic et al., 2022] or [Verwer and Zhang,

 2019] would do. To apply this depth 1 tree to the original continuous attribute dataset, the root node
"$a \in [0.2, 0.22]$" should be decomposed in two decision nodes "$a \leq 0.19$" and "$a \leq 0.22$" before
making a label assignment. So the corresponding tree that can be applied on the continuous attribute
is actually of depth 2.

|       | $a$   | $y$ |       | $[0, 0.19]$ | $[0.2, 0.22]$ | $[0.23, 0.3]$ | $y$ |
|-------|-------|-----|-------|-------------|---------------|---------------|-----|
| $x_1$ | 0.1   | 1   | $x_1$ | 1           | 0             | 0             | 1   |
| $x_2$ | 0.2   | 2   | $x_2$ | 0           | 1             | 0             | 2   |
| $x_3$ | 0.22  | 2   | $x_3$ | 0           | 1             | 0             | 2   |
| $x_4$ | 0.3   | 1   | $x_3$ | 0           | 0             | 1             | 1   |

$\leftarrow$

## F.2 Experiments

Comparing baselines such as [Verwer and Zhang, 2019] or [Demirovic et al., 2022] to DPDT or
Quant-BnB [Mazumder et al., 2022] that operate directly on continuous attributes with the same
maximum depth is not fair in favor of the latter algorithms as discussed above. Still, for the sake of
curiosity we performed comparisons on datasets of prior works. These can be split into two groups.

**1) MurTree:** Demirovic et al. [2022] propose an algorithm that retrieves optimal trees for large
datasets with binary features using dynamic programming. They also propose a binarization method
to retrieve suboptimal shallow trees for large datasets with continuous features. We do not run
MurTree but use of the results in Table 6 from Mazumder et al. [2022] (see the "approx" column)
which previously compared Quant-BnB to MurTree.

**2) OCT, MFOCT, BinOCT:** Bertsimas and Dunn [2017], Aghaei et al. [2020], Verwer and
Zhang [2019] propose optimal tree algorithms which formulate the learning problem as a MIP.
OCT and MFOCT can produce optimal trees for small datasets with continuous features. BinOCT
can also produce optimal trees for small datasets with continuous features after they have been
binarized. We make use of the results available at https://github.com/LucasBoTang/Optimal_
Classification_Trees.

**Reproducibility:** as mentioned above, we did not run the additional baselines but instead used
available results. As such runtimes were provided only when available. OCT, MFOCT, BinOCT were
run on a single core of an Intel(R) Core(TM) CPU i7-7700HQ @ 2.80GHz. MurTree was run
on a single core of a Intel Xeon 2.30GHz. According to online benchmarks the performances of
those machines are similar to our Laptopt CPU Intel® Core™ i7-8665U CPU.

# G Markov Decision Problem formulations of the Decision Tree Learning Problem

In this section we compare our Markov Decision Problem (MDP) formulation of decision tree
learning from Section 4 to that of prior work, namely [Garlapati et al., 2015] and [Topin et al.,
2021]. **In a nutshell**, prior work viewed the task as a deterministic and Partially Observable MDP
[Sigaud and Buffet, 2013] and used algorithms such as Q-learning [Garlapati et al., 2015] or deep
Q-learning [Topin et al., 2021] to solve them in an online fashion one datum from the dataset at a
time. Our approach is different in that it builds a stochastic and fully observable MDP. Our MDP
makes it possible to perform two operations that are critical for DPDT: i) being able to call the
tests generating function which does not operate online but needs full offline access of the dataset
ii) being able to efficiently compute through dynamic programming optimal policies for different
complexity-performance trade-offs, which is critical in practice as our improved training accuracy
compared to greedy methods would otherwise quickly lead to overfitting. High level differences

| Datasets | | | | Accuracy of depth-3 trees | | | | |
|---|---|---|---|---|---|---|---|---|
| Names | Samples | Features | Classes | Opt. | DPDT-5 | DPDT-4 | MurTree | CART |
| avila | 10430 | 10 | 12 | 58.5% | 58.5*% | 58 % | 58.5*% | 53.2% |
| bank | 1097 | 4 | 2 | 98.3% | 98% | 98% | 97.3% | 93.3% |
| bean | 10888 | 16 | 7 | 87.1% | 85.6% | 85% | 86.9% | 77.7% |
| bidding | 5056 | 9 | 2 | 99.3% | 99.3*% | 99.3% | 98.1% | 98.1% |
| eeg | 11984 | 14 | 2 | 70.8% | 70.3% | 70% | 68.8% | 66.6% |
| fault | 1552 | 27 | 7 | 68.2% | 68% | 65.7% | 67.3% | 55.3% |
| htru | 14318 | 8 | 2 | 98.1% | 98% | 98% | 97.9% | 97.9% |
| magic | 15216 | 10 | 2 | 83.1% | 83% | 82.7% | 81.1% | 80.1% |
| occupancy | 8143 | 5 | 2 | 99.4% | 99.4*% | 99.3% | 99.1% | 98.9% |
| page | 4378 | 10 | 5 | 97.1% | 97% | 97% | 96.6% | 96.4% |
| raisin | 720 | 7 | 2 | 89.4% | 88.5% | 88.3% | 87.5% | 86.9% |
| rice | 3048 | 7 | 2 | 93.8% | 93.7% | 93.6% | 93.4% | 93.3% |
| room | 8103 | 16 | 4 | 99.2% | 99.2*% | 99.2% | 99.2*% | 96.8% |
| segment | 1848 | 18 | 7 | 88.7% | 88.2% | 88.2% | 88.1% | 57.4% |
| skin | 196045 | 3 | 2 | 96.9% | 96.7% | 96.7% | 96.8% | 96.6% |
| wilt | 4339 | 5 | 2 | 99.6% | 99.5% | 99.5% | 98.7% | 99.3% |

Table 3: Training accuracy of different decision tree learning algorithms. All algorithms learn trees of depth at most 3 on 16 classification datasets. MurTree returns decision trees for datasets binarized using using the minimum description length principle. Results for MurTree are taken from Tables 2 and 6 from [Mazumder et al., 2022].

| Datasets | | | | Train Accuracy depth-5 | | | | | | Test Accuracy depth-5 | | | | | | Runtime depth-5 | | | | | |
|---|---|---|---|---|---|---|---|---|---|---|---|---|---|---|---|---|---|---|---|---|---|
| Names | Samples | Features | Classes | DPDT-4 | DPDT-5 | OCT | MFOCT | BinOCT | CART | DPDT-4 | DPDT-5 | OCT | MFOCT | BinOCT | CART | DPDT-4 | DPDT-5 | OCT | MFOCT | BinOCT | CART |
| balance-scale | 624 | 4 | 3 | 90.9% | 91.0% | 71.8% | 82.6% | 67.5% | 86.5% | 77.1% | 74.8% | 66.9% | 71.3% | 61.6% | 76.4% | 68.34 | 401.71 | 605.51 | 600.1 | 603.95 | < 0.001 |
| breast-cancer | 276 | 9 | 2 | 94.2% | 94.7% | 88.6% | 91.1% | 75.4% | 87.9% | 66.4% | 67.6% | 67.1% | 73.8% | 62.4% | 70.3% | 19.09 | 62.36 | 603.39 | 600.25 | 603.67 | 0.001 |
| car-evaluation | 1728 | 6 | 4 | 92.2% | 92.2% | 70.1% | 80.4% | 84.0% | 87.1% | 90.3% | 90.3% | 69.5% | 79.8% | 82.3% | 87.1% | 5.39 | 38.07 | 618.09 | 600.49 | 613.14 | < 0.001 |
| hayes-roth | 160 | 9 | 3 | 93.3% | 94.2% | 82.9% | 95.4% | 64.6% | 76.7% | 75.4% | 71.2% | 77.5% | 77.5% | 54.2% | 69.2% | 0.91 | 2.58 | 602.02 | 600.19 | 601.83 | 0.001 |
| house-votes-84 | 232 | 16 | 2 | 100.0% | 100.0% | 100.0% | 100.0% | 100.0% | 99.4% | 95.4% | 95.4% | 93.7% | 94.3% | 96.0% | 95.1% | 0.44 | 0.65 | 105.72 | 10.74 | 6.6 | < 0.001 |
| soybean-small | 46 | 50 | 4 | 100.0% | 100.0% | 100.0% | 100.0% | 76.8% | 100.0% | 93.1% | 93.1% | 94.4% | 91.7% | 72.2% | 93.1% | 0.01 | 0.01 | 4.18 | 0.41 | 1.84 | < 0.001 |
| spect | 266 | 22 | 2 | 93.0% | 93.0% | 92.5% | 93.0% | 92.2% | 88.5% | 73.1% | 73.9% | 75.6% | 74.6% | 73.1% | 75.1% | 6.32 | 16.78 | 604.87 | 600.33 | 605.57 | 0.001 |
| tic-tac-toe | 958 | 24 | 2 | 90.8% | 91.1% | 68.5% | 76.1% | 85.7% | 85.8% | 82.1% | 82.5% | 69.6% | 73.6% | 79.6% | 81.0% | 107.94 | 626.34 | 615.28 | 600.45 | 621.81 | 0.001 |

Table 4: Train/test accuracies and runtimes of different decision tree learning algorithms. Note that we are not using any regularization in this experiment (in order for all solvers to optimize the same objective function) and as such we might overfit compared to CART that does not optimize the training error as intensively. All algorithms learn trees of depth at most 5 on 8 classification datasets. A time limit of 10 minutes is set for OCT-type algorithms. DPDT is used with two different test generating functions: CART with a maximum depth of 4 and CART with a maximum depth of 5. The values in this table are averaged over 3 seeds giving 3 different train/test datasets.

between MDPs are summarized in Table 5. For the sake of self-completeness we then detail both MDPs of [Topin et al., 2021] and [Garlapati et al., 2015] which are to be contrasted with our MDP formulation in Section 4.

Table 5: MDP formulations of the decision tree learning problem

| MDP properties | IBMDP [Topin et al., 2021] | [Garlapati et al., 2015] | Ours |
|---|---|---|---|
| Training samples attributes | Any | Categorical | Any |
| Discounted | Yes | Yes | No |
| Horizon | Infinite | Finite | Finite |
| States | Partial information about a single training sample | Partial information about a single training sample | A full dataset in $\mathcal{P}(\mathcal{D})$ |
| Actions | Tests and label assignments | State dependent tests and label assignments | State dependent tests and label assignments |
| Transitions | Deterministic | Deterministic | Stochastic |

### G.1 Iteratvie Bounding MDPs

An IBMDP [Topin et al., 2021] is an episodic, infinite horizon, discounted MDP. IBMDPs can be used for learning decision trees of any base MDP. We discuss here the case where the base MDP is a classification task. In this case, during each episode, an agent has to classify a hidden training sample $x_i$ drawn uniformly from a training dataset with continuous attributes. We assume whiteout loss of generality that the training dataset $\mathcal{X} \subset [0,1]^{N \times p}$ has continuous attributes in $[0,1]$. On the other hand, the set of labels is $\mathcal{Y} = \{1, ..., K\}$. An IBMDP is defined as follows.

**State space:** the state space is the hypercube $[0,1]^{3 \cdot p}$. A IBMDP state has two parts. The continuous attributes of the hidden training sample $x_i = (x_{i1}, ..., x_{ip})$ to classify, and a lower and upper bound $(L_k, U_k)$ for each of the $p$ attributes. For each attribute $x_{ik}$, $(L_k, U_k)$ represents the current agent knowledge about its hidden value. Initially, $(L_k, U_k) = (0, 1)$ for all $k$, which are iteratively refined by taking tests actions.

**Action space:** an agent in an IBMDP can either take an assignment action $a \in \mathcal{Y}$, or a test action $\mathbb{1}_{\{x_{ik} \leq v \cdot (U_k - L_k) + L_k\}}$ with $k \in \{1, \ldots, p\}$ and $v \in \{\frac{1}{d+1}, ..., \frac{d}{d+1}\}$, with $d \in \mathbb{N}$ a hyperparameter of the IBMDP.

**Transition function:** if an agent takes a label assignment action, the IBMDP transits to a terminal state, a new training sample $x$ is drawn at random from $\mathcal{X}$, and the attributes bounds $(L_1, ..., L_p, U_1, .., U_p)$ are reset to 0 or 1. If an agent takes a test action, the attributes bounds are refined. Let $x_{ik}$ be the value of the k-$th$ attribute of the hidden training sample $x_i$, and $(L_k, U_k)$ be the current bounds of $x_{ik}$. If $\mathbb{1}_{\{x_{ik} \leq v \cdot (U_k - L_k) + L_k\}}$ is true, then $L_k$ is updated to $v \cdot (U_k - L_k) + L_k$, else, it is $U_k$ that is updated to $v \cdot (U_k - L_k) + L_k$.

**Reward function:** the reward for assigning the label $y_i \in \mathcal{Y}$ to the hidden training sample $x_i$ is $\mathbb{1}_{a=y_i} \cdot r_+ + \mathbb{1}_{a \neq y_i} \cdot r_-$, with $r_+ > 0$ and $r_- < 0$. The reward for taking a test action is $\alpha < 0$.

### G.2 MDP formulation of [Garlapati et al., 2015]

MDP formulations based on [Garlapati et al., 2015] assume categorical attributes, i.e, the training dataset $\mathcal{D}$ is in $\mathbb{Z}^{N \times p}$. The MDP is episodic with a discount factor and a finite horizon $p + 1$. An episode of this MDP consists of *costly* queries of a training sample's attributes until a label assignment is made.

**State space:** a state of the above MDP has partial information about a training sample to classify. At every step of the MDP, an agent queries a hidden attribute and updates its knowledge about the training sample by concatenating all revealed attributes.

**Acion space:** at every step $t$ in the MDP, an agent can either assign a class label in $\mathcal{Y} = \{1, ..., K\}$, or, make a query $a_t$ of a hidden attribute of a training sample: $A_t = (\{1, ..., p\} \setminus \bigcup_{h=0}^{t-1} a_h) \cup \mathcal{Y}$.

**Transition function:** the current state of the MDP contains values of previously queried attributes. At $t = 0$, $s = \{\}$. Assuming the hidden training sample to be classified during the current episode is $x_i = (x_{i1}, ..., x_{ip})$, then the deterministic transition function is: $T(s, a = x_{ij}) = s \cup x_{ij}$ or $T(s, a \in \mathcal{Y}) = s_{terminal}$. At the start of a new episode, a new training sample is drawn uniformly from $\mathcal{D}$.

**Reward function:** at time $t$, when the hidden training sample to classify is $x_i$, if the an agent takes an assignment action $a \in \mathcal{D}$, the reward is $\mathbb{1}_{a=y_i} \cdot r_+ + \mathbb{1}_{a \neq y_i} \cdot r_-$, with $r_+ > 0$ and $r_- < 0$. So an agent gets a positive signal for making a correct label assignment and negative signal otherwise. If the agent takes a query action, the reward is a negative value $\alpha$ in order to discourage taking to much queries and control the tree complexity.

## H  Proof of equivalence of learning objectives

In this section, we prove the equivalence between learning an optimal policy in the MDP of Section 4 and finding the minimizing tree of Eq. (2). We first define $C(T)$, the expected number of tests performed by tree $T$ on dataset $\mathcal{D}$. Here $T$ is induced by policy $\pi$, i.e. $T = E(\pi, s_0)$. $C(T)$ can be defined recursively as $C(T) = 0$ if $T$ is a leaf node, and $C(T) = 1 + p_l C(T_l) + p_r C(T_r)$, where $T_l = E(\pi, s_l)$ and $T_r = E(\pi, s_r)$. In words, when the root of $T$ is a test node, the expected number of tests is one plus the expected number of tests of the left and right sub-trees of the root node.

For the purpose of the proof, we overload the definition of $J_\alpha$ and $\mathcal{L}_\alpha$, to make explicit the dependency on the dataset and the maximum depth. As such, $J_\alpha(\pi)$ becomes $J_\alpha(\pi, \mathcal{D}, D)$ and $\mathcal{L}_\alpha(T)$ becomes $\mathcal{L}_\alpha(T, \mathcal{D})$. Let us first show that the relation $J_\alpha(\pi, \mathcal{D}, 0) = -\mathcal{L}_\alpha(T, \mathcal{D})$ is true. If the maximum depth is $D = 0$ then $\pi(s_0)$ is necessarily a class assignment, in which case the expected number of tests is zero and the relation is obviously true since the reward is minus the average classification loss. Now assume it is true for any dataset and tree of depth at most $D$ with $D \geq 0$ and let us prove that it holds for all trees of depth $D + 1$. For a tree $T$ of depth $D + 1$ the root is necessarily a test node. Let $T_l = E(\pi, s_l)$ and $T_r = E(\pi, s_r)$ be the left and right sub-trees of the root node of $T$. Since both sub-trees are of depth at most $D$, the relation holds and we have $J_\alpha(\pi, X_l, D) = \mathcal{L}_\alpha(T_l, X_l)$ and $J_\alpha(\pi, X_r, D) = \mathcal{L}_\alpha(T_r, X_r)$, where $X_l$ and $X_r$ are the datasets of the "right" and "left" states to which the MDP transitions—with probabilities $p_l$ and $p_r$—upon application of $\pi(s_0)$ in $s_0$, as

described in the MDP formulation. Moreover, from the definition of the policy return we have

$$
\begin{aligned}
J_\alpha(\pi, \mathcal{D}, D+1) &= -\alpha + p_l * J_\alpha(\pi, X_l, D) + p_r * J_\alpha(\pi, X_r, D) \\
&= -\alpha - p_l * \mathcal{L}_\alpha(T_l, X_l) - p_r * \mathcal{L}_\alpha(T_r, D) \\
&= -\alpha - p_l * \left( \frac{1}{|X_l|} \sum_{(x_i,y_i) \in X_l} \ell(y_i, T_l(x_i)) + \alpha C(T_l) \right) \\
&\quad - p_r * \left( \frac{1}{|X_r|} \sum_{(x_i,y_i) \in X_r} \ell(y_i, T_r(x_i)) + \alpha C(T_r) \right) \\
&= -\frac{1}{N} \sum_{(x_i,y_i) \in X} \ell(y_i, T(x_i)) - \alpha(1 + p_l C(T_l) + p_r C(T_r)) \\
&= -\mathcal{L}(T, \mathcal{D})
\end{aligned}
$$

# I   Deeper trees experiments

In this section, we push the limits of DPDT to learn trees of at most depth 10. We run two instances of DPDT. The first one will generate a MDP using a depth dependant tests generating function. DPDT-2... generates a MDP where actions availabe at states corresponding to depth $\leq 5$ are given by running CART with a maximum depth of 2, and actions for other states are given by CART with a maximum depth of 1 (the maximum information gain splits given the dataset $X$ in the state $((X, d))$). DPDT-2+1... generates a bigger MDP than DPDT-2... as actions available to states with depths up to 6 are given by CART run with a maximum depth of 2. On Table 6 we observe that deep trees learnt by CART and DPDT perform similarly well on unseen data of different classificiation problems. CART runs way faster than DPDT to compute deep trees. However, DPDT learns more interpretable trees with respect to the average number of tests performed on data which is a very useful feature for real-life applications such as medicine where each additional test before a diagnostic can be very expensive (for example performing an addition MRI scan).

Table 6: Test accuracy of trees of depth $\leq 10$ selected with the procedure described in Sec. 6.2.

| Datasets | Accuracy (%) on unseen data | | | Runtime (s.) | | | Average Nb.Tests | | |
|---|---|---|---|---|---|---|---|---|---|
| Names | DPDT-2... | DPDT-2+1... | CART | DPDT-2... | DPDT-2+1... | CART | DPDT-2... | DPDT-2+1... | CART |
| avila | 94.3 | **95.1** | 87.8 | 86.476 | 187.313 | **1.579** | **8.4** | **8.4** | 8.8 |
| bank | **99.3** | **99.3** | **99.3** | 1.664 | 2.174 | **0.028** | **3.3** | **3.3** | 3.4 |
| bean | **91.3** | 90.9 | 91.2 | 102.796 | 309.981 | **8.287** | 5.2 | **4.0** | 6.1 |
| bidding | **99.4** | **99.4** | **99.4** | 1.833 | 3.226 | **0.095** | **2.4** | **2.4** | **2.4** |
| eeg | **83.6** | 83.5 | 82.0 | 85.198 | 229.49 | **2.386** | **8.1** | 8.2 | 9.3 |
| fault | 73.3 | **73.8** | 68.7 | 35.09 | 108.265 | **1.148** | **5.6** | **5.6** | 6.9 |
| htru | 97.6 | 98.0 | **98.1** | 45.941 | 123.689 | **4.234** | 2.2 | **1.2** | 3.4 |
| magic | **85.4** | 84.9 | 84.8 | 146.253 | 391.594 | **7.021** | **5.8** | 5.9 | 8.1 |
| occupancy | **99.5** | **99.5** | **99.5** | 6.847 | 15.608 | **0.226** | **1.0** | **1.0** | 1.4 |
| page | 96.5 | **96.9** | 96.5 | 22.526 | 58.102 | **0.713** | **4.5** | 6.2 | 7.7 |
| raisin | 85.6 | 86.7 | **88.9** | 8.717 | 19.652 | **0.115** | **2.1** | **2.1** | 6.5 |
| rice | 93.4 | 93.2 | **93.7** | 20.18 | 44.867 | **0.626** | **1.8** | **1.8** | 3.0 |
| room | 99.3 | **99.6** | **99.6** | 5.186 | 8.55 | **0.318** | **2.3** | 4.1 | 4.1 |
| segment | **97.0** | **97.0** | 94.8 | 9.796 | 22.562 | **0.286** | 5.1 | 5.1 | **5.0** |
| skin | **99.9** | **99.9** | 99.8 | 120.576 | 308.577 | **2.94** | 6.3 | 6.2 | **5.4** |
| wilt | **86.0** | **86.0** | 84.8 | 2.274 | 3.583 | **0.151** | **4.3** | 4.4 | 4.4 |

# J   Additional comparisons with Quant-BnB

In Table 7 we compare DPDT with Quant-BnB on train and test sets of different classification problems. Quant-BnB has a time limit equal to DPDT-5' runtime on each problem. We also run Quant-BnB with bonuses of 5 and 50 seconds to see if the latter can outperform DPDT with just a little more time or if it would require almost twice the time (see Table 7 for DPDT-5' runtimes). We observe that for both train and test accuracies, Quant-BnB-t+50 (DPDT-5 runtime plus 50 seconds bonus) outperforms DPDT most often.

Table 7: Train and Tests accuracies of DPDT and Quant-BnB for Trees of maximum depth 3

| Datasets | Train Accuracies | | | | | | Test Accuracies | | | | | |
|---|---|---|---|---|---|---|---|---|---|---|---|---|
| Names | DPDT-3 | DPDT-4 | DPDT-5 | Quant-BnB-$T$ | Quant-BnB-t+5 | Quant-BnB-t+50 | DPDT-3 | DPDT-4 | DPDT-5 | Quant-BnB-$T$ | Quant-BnB-t+5 | Quant-BnB-t+50 |
| avila | 58 | 58 | **58.5** | 57.3 | 57.3 | 57.3 | 57.9 | 57.9 | **58.2** | 57.1 | 57.1 | 57.1 |
| bank | 98 | 98 | 98 | 97.1 | **98.3** | **98.3** | 97.8 | 97.8 | 97.8 | 97.8 | 97.8 | 97.8 |
| bean | 85 | 85 | **85.6** | 85.3 | 85.3 | 85.3 | 85.1 | 85.1 | 84.9 | **85.6** | **85.6** | **85.6** |
| bidding | **99.3** | **99.3** | **99.3** | 98.6 | 98.7 | **99.3** | 99 | **99** | **99** | 98.6 | 98.7 | **99** |
| eeg | 69.4 | 70 | **70.3** | 68.3 | 68.3 | 68.9 | **71** | 69.8 | 70 | 69.8 | 69.8 | 68.5 |
| fault | 65.7 | 65.7 | **68** | 64.6 | 64.6 | 66.9 | 64.8 | 64.8 | **65.3** | 63.2 | 63.2 | 64.3 |
| htru | 98 | 98 | 98 | 98 | 98 | 98 | **98.2** | 97.9 | 97.9 | 98.1 | 98.1 | 98.1 |
| magic | 82.7 | 82.7 | **82.9** | 82.6 | 82.6 | 82.7 | 82 | 82 | 82.2 | 82.2 | 82.2 | **82.3** |
| occupancy | 99.3 | 99.3 | **99.4** | 99.3 | 99.3 | **99.4** | 93.4 | 93.4 | **93.9** | 89.6 | 89.6 | 91 |
| page | **97** | **97** | **97** | 96.5 | 96.7 | **97** | 96 | 96 | 96.1 | **96.7** | 95.9 | 95.9 |
| raisin | 88.3 | 88.3 | 88.5 | 88.1 | 88.6 | 89 | 87.2 | 87.2 | 88.3 | 88.9 | 88.3 | **89.4** |
| rice | 93.5 | 93.6 | **93.7** | **93.7** | **93.7** | **93.7** | 92.1 | 92.1 | **92.7** | 92 | 92 | 92 |
| room | **99.2** | **99.2** | **99.2** | 98.8 | 98.8 | 99 | **99** | **99** | **99** | 98.6 | 98.6 | 98.8 |
| segment | **88.2** | **88.2** | **88.2** | 79.1 | 87.8 | 87.8 | 84 | 84 | 84 | 76.8 | **84.2** | **84.2** |
| skin | 96.7 | 96.7 | 96.7 | 96.7 | 96.7 | 96.7 | **96.7** | **96.7** | **96.7** | 96.6 | 96.6 | 96.6 |
| wilt | 99.5 | 99.5 | 99.5 | 99.4 | 99.4 | **99.6** | 80.4 | 79.2 | 79.2 | 77.6 | **81.2** | 78.8 |

## K    Additional figures for different complexity measures

We show here the complexity-performance trade-offs for all 16 datasets. We show the plot for two complexity measures: average number of tests (what DPDT optimize) and total number of nodes (what the post-process prunning of CART optimizes). On the first measure, the trees that DPDT finds dominate those of CART, which matches the theory. On the second measure, even though we do not optimize for the total number of nodes, we are still able to find better trade-offs w.r.t. this metric than CART for several datasets.

### K.1    Average number of tests vs accuracy

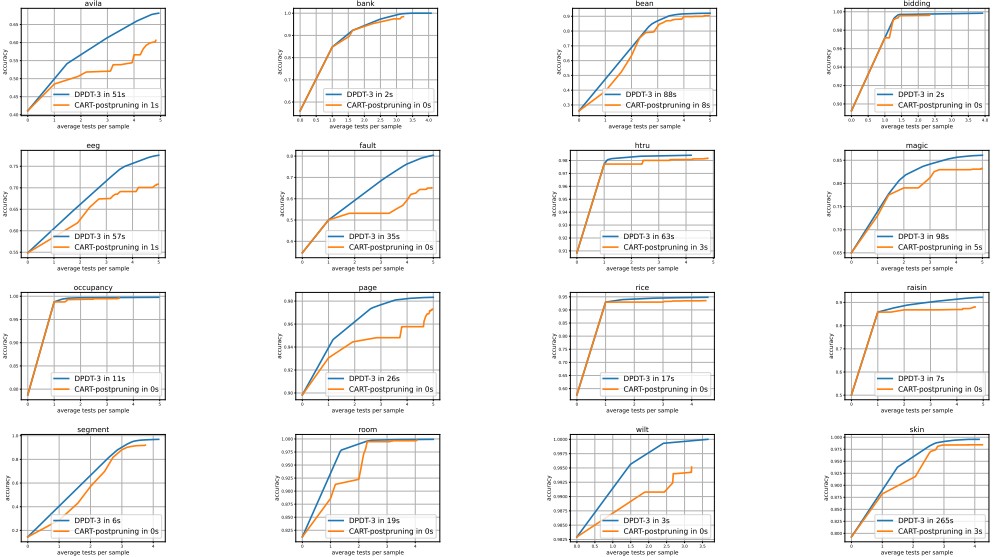

Figure 8: Average number of tests-accuracies trade-offs of CART and DPDT-3 on classification training datasets. Both algorithms learn trees of depths at most 5. CART makes a trade-off with the minimal complexity post-pruning algorithm. DPDT-3 makes a trade-off by returning policies for 1000 different $\alpha$.

### K.2    Total number of nodes vs accuracy

## L    Codes to reproduce experiments

Anonymized github for DPDT code: https://anonymous.4open.science/r/reproduce-E9BD/README.md

Anonymized github of our clone of Quant-BnB code: https://anonymous.4open.science/r/reproduce-quant-bnb-80ED/README.md

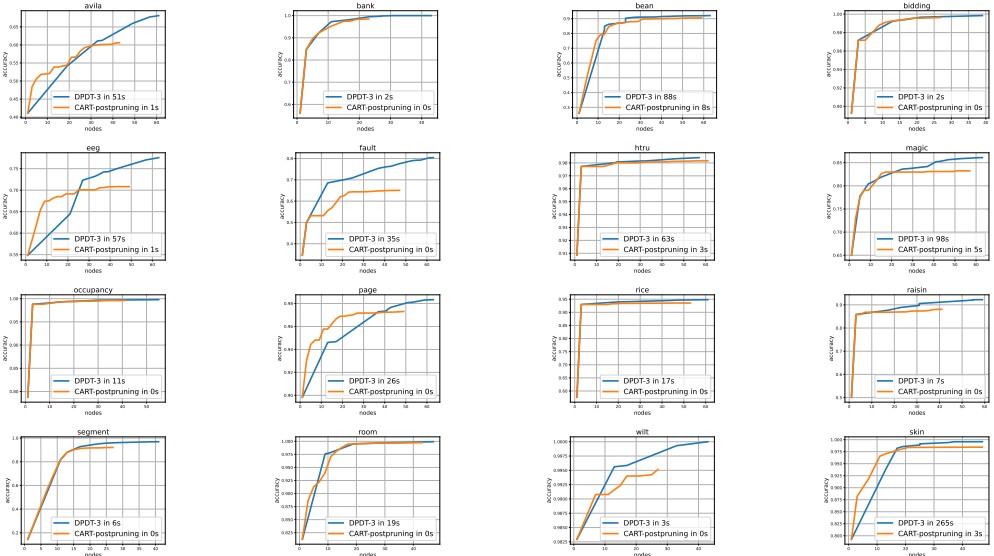

Figure 9: Nodes-accuracies trade-offs of CART and DPDT-3 on classification training datasets. Both algorithms learn trees of depths at most 5. CART makes a trade-off with the minimal complexity post-pruning algorithm. DPDT-3 makes a trade-off by returning policies for 1000 different $\alpha$. Even though we do not optimize for this complexity metric, we are still able to find better trade-offs than CART with post-pruning in several cases.

