# OpenReview forum: "Interpretable Decision Tree Search as a Markov Decision Process"
_NeurIPS.cc/2024/Conference — Submitted to NeurIPS 2024_

### Official Review · Reviewer_mGGd · 2024-07-03

**Soundness:** 2
**Presentation:** 2
**Contribution:** 2
**Rating:** 4
**Confidence:** 4

**Summary:**

This paper formulated the problem of finding an optimal decision tree as Markov Decision Problem and solve the scalability problem using an information-theoretic test generation function. This method provides a trade-off between the train accuracy and tree sizes, the decision tree naturally offers interpretability over ML algorithms.

**Strengths:**

This paper is well-written and well-organized, it combines RL and decision tree generation, and building MDP before constructing the decision tree.

**Weaknesses:**

1. The paper lacks sufficient novelty. The approach of constructing a Markov Decision Process (MDP) and using Decision Trees (DT) to generate actions is not new. Specifically, Algorithm 1 appears to still rely on Classification and Regression Trees (CART) for splitting criteria, which diminishes the originality of the proposed method.
2. The evaluated scenarios in the paper are not clearly articulated. The algorithm has not been tested against well-known benchmarks, unlike other optimal DT algorithms. This makes it difficult to assess the comparative performance and robustness of the proposed approach.
3. The advantages of using this algorithm instead of CART are not clearly demonstrated. Both algorithms control tree size and depth. However, CART is known to converge faster and offers a simpler implementation. Without clear evidence of the benefits, it is hard to justify the use of the proposed method over established techniques like CART.
4. The definition of actions generated by the tree is ambiguous. It is not clear whether the actions are discrete or continuous. If the algorithm is designed to build an MDP, it should be tested on general reinforcement learning (RL) tasks to validate its effectiveness and applicability in broader contexts.

**Questions:**

See weakness section above.

**Limitations:**

Yes

---

> ### Author Rebuttal · Authors · 2024-08-01
>
> Dear reviewer,
>
> Thank you in advance for engaging in a discussion with us. We appreciate your remarks that show you have studied our work well; thank you!
>
> $\textbf{Novelty of DPDT}$
>
> Our MDP formulation of decision tree learning is the first to be applicable to both continuous and categorical data attributes. We compare existing MDP formulations to our approach (DPDT) in Appendix G. Although Algorithm 1 still relies on CART, we believe this is where the beauty and simplicity of our work is best demonstrated. Let us explain.
> The spectrum of decision tree algorithms has two ends. On one end, there is CART the well-known heuristic that chooses how to partition the training data by only looking at the current entropy maxmization in the targets. CART does not consider how the first split chosen will affect the overall performance of the future splits. On the other end optimal decision tree algorithms such as Quant-BnB to which we compare ourselves computes all possible trees (combination of splits) and returns the one with highest accuracy.
> DPDT can be anywhere on that spectrum. DPDT can return the same tree than CART by setting $K=1$ in algorithm 1. DPDT can also return the same tree than optimal algorithm such as Quant-BnB By replacing splits returned by CART in algorithm 1 by the set of all possible splits. The key thing is that considering all splits is costly (combinatorial optimization problem). So by not considering all splits (optimal algorithms) and by not considering only single splits (CART) but by considering some small set of splits such as the ones from a heuristic tree obtained with CART; DPDT is somewhere in the middle of the spectrum of algorithms with accuracy close to the optimal trees and has runtimes order of magnitudes shorter than optimal algorithms. Surprisingly, we showed that DPDT can approach QuantBnB performance (table 1) despite exploring a significantly smaller susbet of splits (figure 1).
> The other novely of our MDP formulation is that it allows for the computation of many trees at the same time for the training data but different regularization values (see section 5.2 and figure 3)!
>
> $\textbf{Advantages of DPDT over CART}$
>
> Indeed, CART is a simple heuristic that works fine in practice for generalization tasks. Our work, DPDT, offers advantages over CART when a user has interpretability or cost constraints. By definition of algorithm 1, DPDT trees will always have shorter decision path in average than CART trees for equal accuracies. In parctice we show this in Fig 2 right plot where we bar plot the gain over CART. Similarly, it is possible to give feature costs as input for DPDT, then the resulting trees will trade-off automatically between feature cost and accuracy. We show in a new experiment that DPDT pareto dominates CART trees in practice for this application too: see global rebuttal (1-page pdf at top of open review page). A real-life use of feature cost could be when testing a feature is more expensive than others, e.g, testing with an M.R.I is more expensive than testing the size of a patient. DPDT optimizes cost-accuracy or decision length-accuracy naturally trought the MDP reward.
>
> $\textbf{Optimal decision trees benchmarks}$
>
> The datasets on which we test DPDT are common to recent optimal tree litterature such as [1, table 1], [2, table 4] and [2, table 2]. Please let us know if there are specifics dataset on which you like us to test DPDT.
>
> We do not understant the 4 th weakness mentioned by the reviewer but we are curious to hear more.
>
> $\textbf{Conclusion}$
>
> In conclusion we believe that figure 2 and 3 already demonstrates the advantage of DPDT over CART in a context of interpretability. Would the reviewer consider raising their score if we also include an experiment that demonstrates DPDT superiority over CART for a task with feature costs?
>
> Thank you in advance.
>
> [1] MurTree: Optimal Decision Trees via Dynamic Programming and Search,  Demirovic et. al., JMLR 2022
>
> [2] Blossom: an Anytime Algorithm for Computing Optimal Decision Trees,  Demirovic et. al., ICML 2023
>
> [3] Quant-BnB:  Scalable Branch-and-Bound Method for Optimal Decision Trees with Continuous Features, Rahul Mazumder et.al., ICML 2022

---

> > ### Comment · Reviewer_mGGd · 2024-08-09
> >
> > Thank you for answering my questions, I have read the authors' submitted rebuttal, and I would like to maintain my score.

---

> > > ### Author Response · Authors · 2024-08-12
> > > **Question**
> > >
> > > Dear reviewer,
> > >
> > > What are the benchmarks you had in mind when you said "The evaluated scenarios in the paper are not clearly articulated. The algorithm has not been tested against well-known benchmarks, unlike other optimal DT algorithms. This makes it difficult to assess the comparative performance and robustness of the proposed approach." ?
> > >
> > > Furthermore, how can we convince you that our approach has many advantages over CART in addition to the what we wrote in our rebuttal?
> > >
> > > Thank you

---

> ### Author Response · Authors · 2024-08-08
> **Rebuttal**
>
> Dear reviewer,
>
> Please take the time to read our rebuttal, we would really appreciate it.
>
> Thank you in advance

---

### Official Review · Reviewer_pDp5 · 2024-07-07

**Soundness:** 3
**Presentation:** 3
**Contribution:** 2
**Rating:** 6
**Confidence:** 3

**Summary:**

The authors pose binary decision tree construction within the framework of Markov Decision Processes. They first propose methods for constructing an MDP from a decision tree construction problem, exploring varying test generating functions that trade off the coverage of the search space vs the size of the search space. They then apply Dynamic programming to solve the resulting MDP and show this learnt method can both create binary trees that minimise the loss over a dataset but that it can also be used to add additional losses a user may have over decision trees such as the prior that trees should be small, making them interpretable. They evaluate their proposed method, comparing with other high performing methods such as Quant-BnB, MurTree and a DeepRL method.

**Strengths:**

- Well written paper and easy to understand the method
- Clearly an important direction of research
- Thorough experiments with appropriate baselines and good range of datasets to ensure the conclusions generalise to a wide range of datasets
- Code fully provided, along with implementations of baselines used

**Weaknesses:**

- Various versions of Reinforcement Learning for binary tree construction have previously been explored. While the implementation in this paper is ultimately different and appears to significantly improve performance, there is limited novelty of the approach. Novelty largely comes down to the test generating functions explored and the addition of extra losses (interpretability) in addition to just the dataset accuracy.

- Small formatting issue
     - Table 1 is too small

**Questions:**

- Do you have any intuition as to why Deep Reinforcement Learning methods fail in this domain, compared to non Deep RL approaches. And do these reasons present any problems when scaling this method to larger datasets where neural learning would perhaps be introduced.

**Limitations:**

- Limitations are appropriate addressed

---

> ### Author Rebuttal · Authors · 2024-08-05
>
> Dear reviewer,
>
> We thank you so much for your review. Your commments really reflect your inverstment in reading and understanding our work and we are very excited to engage in a discussion with you!
>
> $\textbf{DPDT has many advantages over other MDP formulations}$
>
> It is true that various RL approaches already exist for constructing decision trees. As the reviewer pointed out novel MDP formulation allows for user-defined costs such as costs of adding nodes to control the complexity (interpretability) of the tree. By definition, other MDP formulations of decision tree construction might already control the tree complexity with the MDP discount factor. In the following we highlight how our MDP formulation and solver DPDT are unique and promising for constructing decision trees in real use cases.
>
> - $\textit{Interpretabillity is not the only cost that DPDT can deal with}$. Indeed, other costs such as feature costs can be used as the reward in the MDP construction. We added an example in the global 1 page pdf rebuttal on top of the Open Review page where some features of the training data are more expensive than other to test in the decsision process. We show that compared to the widely used in practice CART, DPDT can construct trees that better trade-off between feature costs and accuracy. Other MDP formulations and RL work tackled this problem too [1] but cannot be used in practice the same way as DPDT. We explain in the second point why and highlight another promising novelty of DPDT.
>
> - $\textit{Unlike any other MDP forumlation, DPDT computes a whole pareto front of trees at once.}$ Other MDP formulations [1, 2] that include a cost for interpretability or features in the MDP reward, DPDT solves the Bellman optimality equation for multiple costs at once (see section 5.2 of our work). This is a strong feature of DPDT as a user would not have to worry about what cost to choose for interpretability and re-run algorithms [1] or [2] if they are not satisfied with the resulting tree cost-accuracy trade-off: DPDT returns all the trees on the cost-accuracy pareto front and the user can the choose the one that best suit their need! We believe this is key novelty compared to other MDP formulations. However we aknowledge that indeed solving a set of Bellman optimality equations at once is possible for DPDT because there is no neural learning unlike [1] and [2]. This leads us to the other question of the reviewer regarding scaling with Deep RL.
>
> - $\textit{DPDT has guarantees on accuracy thanks to our test generating functions}$. Unlike other MDP formulations that consider naïve test generating functions [2, sec 4.1, paragraph "Action Space"], we use the CART heuristic to generate tests. Those test are still heuristic but allow for DPDT to have a lower bound on train accuracy that is the train accuracy of CART (see our propostion 2).
>
> $\textbf{Failure modes of Deep RL and scaling DPDT with neural learning}$
>
> We believe scaling DPDT to bigger datasets with neural learning is a promising research avenue aspointed out by the reviewer. For that, existing failure modes of Deep RL should be overcome. There exists a study of failure mode of the Deep RL baseline [1] we use in our work, it is [3]. The main failure mode is as follows. Deep RL of decision trees rely on a neural network learning to predict a feature to test against a value given as input dataset features bounds. Deep learning is used for tasks where the predictions of the neural network should be "similar" for neural net inputs that are "close" from a metric perspective. In our case, the neural net might have to learn to predict a test $f_1$ for data that have feature $x_i \leq 0.5$ and an other test $f_2$ for data that have feature $x_i > 0.5$. What [3] suggests is that it is too hard for Deep RL to learn different predictions for e.g. point (0.2, 0.2, ... 0.2, 0.49, 0.2, ..., 0.2) and point (0.2, 0.2, ... 0.2, 0.51, 0.2, ..., 0.2). So essentially Deep RL cannot fit non-continuous decisions in that context, and the neural architectures used in Deep RL do not generalize properly between data feature bounds.
>
> More specifically; the representation of the state (feature bounds) limits the generalization of the neural architecture: we want to learn a neural network that returns an optimal data split given a dataset as input but all that the neural network receives as input is a bounding box around the dataset. If we want to improve generalization, we need to take into account specificities of the problem, e.g. that an optimal split, up to translation of the threshold, remains optimal if all the data is translated. This requires receiving the whole dataset as input and not only a bounding box of where the data is located
>
> A promising research avenue that would be worth discussing at a top ML conference would be to design neural architecture tailored for test predictions given a whole dataset as input such as [4].
>
> $\textbf{Conclusion}$
>
> We highlighted the advantages of DPDT compared to other MDP formulation, in particular outputing a whole pareto front of trees. The reviwer question also raised a promising research avenue that would be worht discussing in NeurIPS.
> If the esteemed reviewer appreciate our effort to address its concern, we encoure them to raise their score. We are happy to engage in further discussion.
>
> [1] Janisch, J., Pevný, T., & Lisý, V. (2019). Classification with Costly Features Using Deep Reinforcement Learning. Proceedings of the AAAI Conference on Artificial Intelligence.
>
> [2] Topin, Nicholay, et al. (2021). Iterative bounding mdps: Learning interpretable policies via non-interpretable methods. Proceedings of the AAAI Conference on Artificial Intelligence.
>
> [3] Kohler, Hector, et al. (2023). Limits of Actor-Critic Algorithms for Decision Tree Policies Learning in IBMDPs, arXiv 2309.13365.
>
> [4] Kossen et. al. Self-Attention Between Datapoints: Going Beyond Individual Input-Output Pairs in Deep Learning NeurIPS 2021

---

> > ### Comment · Reviewer_pDp5 · 2024-08-08
> >
> > I thank the authors for their thorough rebuttal and am satisfied with the response. I have read the other reviews and responses. I have no further questions and would like to maintain my score.

---

> > > ### Author Response · Authors · 2024-08-08
> > > **Thank you**
> > >
> > > Thank you

---

> ### Author Response · Authors · 2024-08-08
> **Rebuttal**
>
> Dear reviewer,
>
> Please take the time to read our rebuttal, we would really appreciate it.
>
> Thank you in advance

---

### Official Review · Reviewer_aAr3 · 2024-07-13

**Soundness:** 2
**Presentation:** 2
**Contribution:** 2
**Rating:** 5
**Confidence:** 3

**Summary:**

This paper models the construction of decision trees as a reinforcement learning problem. Currently SOTA algorithms for constructing decision trees have the drawbacks that 1) they take long to compute at depths > 3, and 2) the trees constructed are complex and difficult to interpret. By modelling the problem as an RL task, the authors hope to make the construction of decision trees scale to larger sizes. They present Dynamic Programming Decision Trees which models tree construction as a MDP solved using dynamic programming. They evaluate the accuracy of trees produced by their approach empirically against other commonly used approaches.

**Strengths:**

**Originality:** The approach presented is a novel approach to constructing dynamic trees.

**Significance:** As datasets become larger and interpretability becomes more important, having an approach that scales DT construction to larger trees is needed now. This makes this work rather significant.

**Clarity:** The first half of the paper (up to Sec. 4) was clear. It becomes harder to understand after. Providing an intuitive explanation certain equations would be helpful. E.g., why is probability $p_l = |X_l| / |X|$?

**Quality:** The technique designed is sound and the experiments chosen were the correct ones to demonstrate their claims.

**Weaknesses:**

The experimental evaluation is weak. From what I understood, the algorithms being evaluated were run only once and evaluated once. The results do not statistically back up the authors' claims. Multiple runs with statistical significance testing is needed.

There is no actual analysis on the interpretability of the trees produced, only the complexity of the trees.

**Questions:**

My main concern is the experimental evaluation.

**Limitations:**

The authors mention one limitation (test generation) as a problem. It seems that that would make it difficult for DPDT to actually scale to larger trees. Is that not so? Is scalability not a limitation then?

---

> ### Author Rebuttal · Authors · 2024-07-31
>
> Dear reviewer, thank you for your comments. We are surprised by the low rating given your positive feedbacks. Please engage with us in the following discussion so that we can convince you to raise your score and accept our work.
>
> $\textbf{Clarifying section 4: formulating decision tree learning as a Markov decision process}$
>
> If the reviewer is not familier with MDPs we recommend Sutton and Barto 2018 as an introduction. Now, constructing a decsion tree can be seen as sequentially splitting a dataset $X$ with $n$ samples and $p$ features. At time $t$ an action to split is taken e.g. if the $i$-th sample has feature $j$ less than some value $v$ (a node in a decision tree does this), then at time $t+1$ the dataest to be considered is made of all the samples from $X$ that verify this condition and is noted $X_l$ else, at time $t+1$ the dataest to be considered is made of all the samples from $X$ that do not verify this condition and is noted $X_r$. So when applying the test is feature $j$ less than some value $v$ on a random datum from $X$, the probability to be in $X_l$ is indeed $p_l=|X_l|/|X|$.
>
> We also refer the reviewer to our appendix D for explanatory schematics.
>
> $\textbf{Statistical significance of experimental results}$
>
> We confirm to the reviewer that table 1 presents results obtained from a single seed. This is because we do not consider a train-test generalization task in this table but only compare ourselves to a deterministic baseline Quant-BnB [1] which itselfs present results on a single seed (see [1, table 2]). Furthermore, our algorithm DPDT, Quant-BnB, and CART are deterministic. The only stochastic baselines are Custard which we run on multiple seeds (see table 1).
>
> To showcase our goodwill and for the reviewer to raise its score, we re-do both plots from figure 3 on 5 different train/test splits chosen at random. We present the results in the 1-page general rebuttal pdf (top of the page). Adding multiple seeds does not change our results and adds significance. Thanks to the reviewer for the recommendation.
>
> $\textbf{Interpretability analysis}$
>
> Interpretability of decision trees can have different meanings. We confirm to the reviewer that the deifinition we use in our work is closely related to the complexity of the learned decision trees, i.e. how many operations to go from input to decision. This notion is called simulatability in the explainable machine learning litterature [1] and draws a parallel between computation complexity of model inference and human user understanding of the tree decisions.
>
> What we show in our work is that in average the trees returned by our method DPDT have shorter decision path in average than trees returned by CART (the heursitic baseline widely used in practice). It means that to make a decision given a datum, trees returned by DPDT will perform less operation in average! This what is shown in figure 2 (right plot) and figure 3. In particular, in figure 2 on the right we show that on only 1 out of 16 dataset DPDT trees are less interpretable than CART.
>
>
> $\textbf{Scalability}$
>
> We already showed that DPDT is able to return deep trees (see table 6 appendix I). DPDT deep trees have better test accuracies and shorter average decision paths than CART deep trees on most of the benchmarked datasets! In practice, since DPDT cannot explore as many splits per state when building the MDP (see our algo 1), if one needs to learn deep trees, we recommend exploring less splits closer to the root with the rationale that it is better to be greedy closer to the leafs than the root: the provided code already supports this feature and was used to get table 6.
>
> $\textbf{Conclusion}$
>
> Having addresssed the concerns of the reviewer and having performed the multiple seeds additional experiments, we kindly ask the reviewer to accept our paper or to engage in the rebuttal.
>
> Respectfully.
>
> [1] The Mythos of Model Interpretability, Zachary C. Lipton ICML 2016
>
> [2] See 1-page pdf for the global rebuttal available at the top of the open review page.

---

> ### Author Response · Authors · 2024-08-08
> **Rebuttal**
>
> Dear reviewer,
>
> Please take the time to read our rebuttal, we would really appreciate it.
>
> Thank you in advance

---

> ### Author Response · Authors · 2024-08-09
> **Discussion**
>
> Dear reviewer we would to discuss your review. Did you have time to read our rebuttal?
>
> Thank you in advance!

---

> ### Author Response · Authors · 2024-08-12
> **Discussion period is almost over**
>
> Dear reviewer, please read our rebuttal. We really enjoyed your feedback! Thank you in advance.

---

> > ### Comment · Reviewer_aAr3 · 2024-08-12
> > **Response to rebuttal**
> >
> > The authors do address my main concern about statistical significance. I appreciate the plots with different dataset splits and this helps to strengthen their results.
> >
> > However, a couple things of note:
> > 1. By clarifying Section 4, I think the authors should rewrite or restructure the section so that it is clearer.
> > 2. I do not believe that the authors can claim that the trees produced are more interpretable. The most they can claim from the results is that trees are less complex.
> >
> > Given the response and new results, I will increase my score.

---

> > > ### Author Response · Authors · 2024-08-12
> > > **Response to response**
> > >
> > > Thank you for raising your score! Indeed in an ideal world we would do a user study to evaluate the interpretability of our trees; however the latter being costly, complexity is the best and most justified proxy we found.
> > >
> > > We will clarify section 4.
> > >
> > > Thank you for engaging in the discussion.

---

### Official Review · Reviewer_E4uJ · 2024-07-13

**Soundness:** 3
**Presentation:** 3
**Contribution:** 1
**Rating:** 3
**Confidence:** 5

**Summary:**

The paper proposes to use an approach for learning interpretable
decision trees using markov decision processes. The results are shown
to be competitive with branch and bound methods.

**Strengths:**

None of notice, given the listed weaknesses.

**Weaknesses:**

There exists extensive experimental evidence challenging the claims
about the interpretability of decision trees, while simultaneously
demonstrating the need for decision trees to be explained, since these
can otherwise exhibit arbitrary explanation redundancy.

As a result, and at present, there is no practical justification
whatsoever to learn so-called interpretable optimal decision trees.
It is absolutely unclear that optimal decision trees will provide any
advantage, regarding computed explanations, over decision trees
induced with heuristic algorithms.

Given the above, I cannot recommend acceptance.

Some references on the necessity of explaining decision trees.

Xuanxiang Huang, Yacine Izza, Alexey Ignatiev, João Marques-Silva: On
Efficiently Explaining Graph-Based Classifiers. KR 2021: 356-367

Gilles Audemard, Steve Bellart, Louenas Bounia, Frédéric Koriche,
Jean-Marie Lagniez, Pierre Marquis: On the Computational
Intelligibility of Boolean Classifiers. KR 2021: 74-86

Yacine Izza, Alexey Ignatiev, João Marques-Silva: On Tackling
Explanation Redundancy in Decision Trees. J. Artif. Intell. Res. 75:
261-321 (2022)

Gilles Audemard, Steve Bellart, Louenas Bounia, Frédéric Koriche,
Jean-Marie Lagniez, Pierre Marquis: On the explanatory power of
Boolean decision trees. Data Knowl. Eng. 142: 102088 (2022)

Gilles Audemard, Steve Bellart, Louenas Bounia, Frédéric Koriche,
Jean-Marie Lagniez, Pierre Marquis: On Preferred Abductive
Explanations for Decision Trees and Random Forests. IJCAI 2022:
643-650

João Marques-Silva, Alexey Ignatiev: No silver bullet: interpretable
ML models must be explained. Frontiers Artif. Intell. 6 (2023)

**Questions:**

NOne.

**Limitations:**

These were listed above. I believe the paper is solving a non-relevant
problem given practical and theoretical evidence regarding the
non-interpretability of decision trees, be these optimal or not.

---

> ### Author Rebuttal · Authors · 2024-07-31
>
> Dear reviewer, we argue that learning decision trees that are interpretable in the simulatability sense (the ability for a human to read the decision path of a model from input to decision) [1, sec. 3.1.1], is a very relevant problem in machine learning. Indeed many recent works present decision tree algorithms as interpretable machine learning solutions whereas in reality their classifiers or policies [2,3] have many decision nodes hindering human readability [1, sec. 3.1.1].
>
> Now assuming the problem we tackle is relevant, please note that in our work we do provide evidence that our proposed method learns trees that have shorter decision paths in average than trees returned by purely heuristic methods for the same accuracy. See figures 2 and 3 from our work.
>
> If you are open to discussion, please let us know how we could convince you that are work offers an original competitve solution to the problem of learning interpretable decision trees.
>
> [1] The Mythos of Model Interpretability, Zachary C. Lipton ICML 2016
>
> [2] Oblique Decision Trees from Derivatives of ReLU Networks, Guang-He Lee, Tommi S. Jaakkola ICLR 2020
>
> [3] Verifiable Reinforcement Learning via Policy Extraction, Osbert Bastani, Yewen Pu, Armando Solar-Lezama NeurIPS 2018

---

> ### Author Response · Authors · 2024-08-08
> **Rebuttal**
>
> Dear reviewer,
>
> Please take the time to read our rebuttal, we would really appreciate it.
>
> Thank you in advance

---

> ### Author Response · Authors · 2024-08-09
> **Discussion**
>
> Dear reviewer, we would liove to discuss. Did you have time to read our rebuttal?
>
> Thank you in advance!

---

> > ### Comment · Reviewer_E4uJ · 2024-08-13
> > **Reply to authors**
> >
> > Thank you for the constructive review.
> >
> > I maintain my assessment. Interpretability is widely accepted to be a subjective topic. Arguing for the interpretability of decision trees would always be open to debate. Existing evidence suggests that attempts to learning interpretable decision trees should be expected to unproductive efforts. Assessment of the redundancy of explanations in the computed decision trees should be discussed, and it is not.

---

> ### Author Response · Authors · 2024-08-13
> **Interpretability is not our only contribution**
>
> Dear reviewer, hello again and thank for pointing out references for the inspection of trees explanations.
>
> We will highlight a discussion about the subjectivity of interpretability with respect to some existing definitions [1,2].
> We will also include the pointed out references [3,4,5,6,7,8] and highlight that tree models might need further inspections to be considered interpretable.
>
> However we would really appreciate if the reviewer could give feedback about the other contributions of our work:
>
> - What do you think about our MDP formulation of decision tree learning? You can check out this discussion with reviewer pDp5 https://openreview.net/forum?id=TKozKEMKiw&noteId=EslLWlsjtK
>
> - What do you think of our solver DPDT that can return a whole set of trees way faster than other optimal tree baselines?
>
> Thank you in advance
>
> [1] Lipton. "The mythos of model interpretability." In ICML Workshop on Human Interpretability in Machine Learning, 2016.
>
> [2] Shen. "Interpretability in ml: A broad overview." 2020
>
> [3] Xuanxiang Huang, Yacine Izza, Alexey Ignatiev, João Marques-Silva: On Efficiently Explaining Graph-Based Classifiers. KR 2021: 356-367
>
> [4] Gilles Audemard, Steve Bellart, Louenas Bounia, Frédéric Koriche, Jean-Marie Lagniez, Pierre Marquis: On the Computational Intelligibility of Boolean Classifiers. KR 2021: 74-86
>
> [5] Yacine Izza, Alexey Ignatiev, João Marques-Silva: On Tackling Explanation Redundancy in Decision Trees. J. Artif. Intell. Res. 75: 261-321 (2022)
>
> [6] Gilles Audemard, Steve Bellart, Louenas Bounia, Frédéric Koriche, Jean-Marie Lagniez, Pierre Marquis: On the explanatory power of Boolean decision trees. Data Knowl. Eng. 142: 102088 (2022)
>
> [7] Gilles Audemard, Steve Bellart, Louenas Bounia, Frédéric Koriche, Jean-Marie Lagniez, Pierre Marquis: On Preferred Abductive Explanations for Decision Trees and Random Forests. IJCAI 2022: 643-650
>
> [8] João Marques-Silva, Alexey Ignatiev: No silver bullet: interpretable ML models must be explained. Frontiers Artif. Intell. 6 (2023)

---

> > ### Comment · Reviewer_E4uJ · 2024-08-14
> > **Reply to Authors**
> >
> > I believe the questions posed by the authors miss the whole point of my review and subsequent comments. One can always envision great solutions for problems of arguable relevancy. However, and in the reviewer's opinion, those should not be the focus on papers presented at top-tier conferences.
> >
> > The main issue raised by my comment is not about interpretability, but the fact that the best possible decision trees that one can construct will almost surely require being explained. Also, the size of the tree may matter very little to the computed explanations, and that is a topic of research per se.

---

> > > ### Author Response · Authors · 2024-08-14
> > > **Recent work contradicts the reviewer.**
> > >
> > > Dear reviewer,
> > >
> > > In hope to convince you of the relevance of the problem we solve, namely learning decision trees, we provide additional references supporting interpretability of decision trees as machine learning models.
> > >
> > > [1,2] describe decision trees as the "quintesential" interpretable model class. [1] develops the first theoretical framework for the approximation capabilities of trees with respect to their interpretability. This theory supports our contributions and especially section 6.2 and 6.3 where we study the interprerability performance trade-off of trees.
> > >
> > > Please, could you either raise your score or explain why think learning decision tree and claiming interpretability is false?
> > >
> > > Thank you in advance
> > >
> > > [1] A Theory of Interpretable Approximations
> > > Marco Bressan, Nicolò Cesa-Bianchi COLT2024
> > >
> > > [2] Christoph Molnar. Interpretable Machine Learning. 2 edition, 2022. URL
> > > https://christophm.github.io/interpretable-ml-book.

---

> > > > ### Comment · Reviewer_E4uJ · 2024-08-14
> > > > **Reply to authors**
> > > >
> > > > Since the early 2000s that authors have claimed that decision trees are interpretable. Since interpretability is subjective, this claim has never been subjected to rigorous assessment. One of the cited references proves that explanations redundancy in decision trees can be arbitrarily large on the number of features. That is a rigorous statement, with a supporting proof. More importantly, the theoretical result is supported by some initial experiments, that confirm the same observation. If light of those results, learning optimal decision trees cannot be justified on the grounds of interpretability. Since the paper overlooks these results, and the related bibliography, since the paper does not assess whether or not optimality helps in anyway the explanations of decision trees, I cannot recommend acceptance, and I will stick to the score given. As stated in the review, and in the already many answers to the authors, the score is supported by scientific evidence that the problem being solved is, at present, a non-relevant problem.

---

> > > > > ### Author Response · Authors · 2024-08-14
> > > > > **Discussion**
> > > > >
> > > > > Well, first of all we appreciate you engaging in the discussion even though we disagree.
> > > > >
> > > > > Please keep in mind that we do not claim that optimality helps explainability/interpretability. What we claim is that formulating decision tree learning as solving MDPs is a better heuristic than CART. We show that trees learned by our method DPDT, have less splits in average for better accuracy than CART trees (see pareto fronts in our paper). Furthermore, we show that DPDT can learn trees that have accuracies closer to optimal DTs than CART.
> > > > >
> > > > > Please consider assessing our work with respect to whether or not we support our claims.
> > > > >
> > > > > Respectfully

---

### Author Rebuttal · Authors · 2024-07-31

Dear all, in addition to the attached 1-page pdf containing additional plots to showcase the superiority of DPDT over CART in terms of tree interpretability, with multiple seeds, we would like to share with you an open source implementation of DPDT that fits the scikit-learn framework in an effort of reproducibility and open science: https://anonymous.4open.science/r/dpdt-py-DB89/README.md

---

### Decision · Program_Chairs · 2024-09-25

**Decision:**

Reject

**Comment:**

There was a robust back-and-forth between the authors and reviewers during the rebuttal, which has been carefully considered.

Given the varying evaluations of the paper, the AC also read the paper to inform the final decision.

Thanks everyone for your reviews and engaging with the authors. There are some different opinions here. Overall there are two major weaknesses that have been raised.

Weakness 1: Significance.  Reviewer E4uJ raised the question of whether this is a relevant problem in the sense that small trees are not necessarily more interpretable. It is a weakness that the paper doesn't attempt to evaluate interpretability in any way other than using size as a proxy. Given that there has been decades of decision-tree research, including work on finding small trees and interpretability, the "significance bar" for work on interpretability is quite high.

Weakness 2: Novelty. There were also concerns about novelty. After reviewing the approach, the AC also did not see significant novelty. The approach literally create a depth D expectimax tree and applies dynamic programming. As the authors mention in related work, this approach is like [Nunes et al., 2020] but expanded to deal with a regularized objective. That seems to be a relatively minor change. The poor scalability of the approach is still an issue, which requires significant depth and test-per-node limits.

Strength: Simplicity. On the positive side this is a simple method and I don't want to discount the contribution of identifying a simple approach that can produce solutions for a large set of accuracy-cost trade-offs at one time (seems to be linear in the number of trade-offs considered). Of course the simplicity is enabled by heuristic choices and rather drastic restrictions on depth and tests-per-node.

Overall, the strengths did not outweigh the weaknesses. The decision might be different if either 1) the paper could provide more direct and compelling results for interpretability, or 2) the level of technical innovation in the direction of small trees was larger.